# A local-to-global emissions inventory of macroplastic pollution

Joshua W. Cottom[1], Ed Cook[1] & Costas A. Velis[1✉]

Negotiations for a global treaty on plastic pollution[1] will shape future policies on plastics production, use and waste management. Its parties will benefit from a high-resolution baseline of waste flows and plastic emission sources to enable identification of pollution hotspots and their causes[2]. Nationally aggregated waste management data can be distributed to smaller scales to identify generalized points of plastic accumulation and source phenomena[3–11]. However, it is challenging to use this type of spatial allocation to assess the conditions under which emissions take place[12,13]. Here we develop a global macroplastic pollution emissions inventory by combining conceptual modelling of emission mechanisms with measurable activity data. We define emissions as materials that have moved from the managed or mismanaged system (controlled or contained state) to the unmanaged system (uncontrolled or uncontained state—the environment). Using machine learning and probabilistic material flow analysis, we identify emission hotspots across 50,702 municipalities worldwide from five land-based plastic waste emission sources. We estimate global plastic waste emissions at 52.1 [48.3–56.3] million metric tonnes (Mt) per year, with approximately 57% wt. and 43% wt. open burned and unburned debris, respectively. Littering is the largest emission source in the Global North, whereas uncollected waste is the dominant emissions source across the Global South. We suggest that our findings can help inform treaty negotiations and develop national and sub-national waste management action plans and source inventories.

Plastic pollution is a global challenge requiring immediate action owing its environmental persistence and negative impact on ecosystems[14], infrastructure[15], society and the economy[16]. The importance of this burgeoning issue has recently been recognized by the ratification of a United Nations draft resolution to create an internationally legally binding instrument to end plastic pollution[1], hereafter the 'Plastics Treaty'. A global plastic pollution emissions inventory has been suggested as being critical to the success of the Plastics Treaty[17] and such inventories have already been applied in the climate change field[18] and as early evidence for a global legally binding agreement on mercury[19,20]—eventually the Minamata Convention[21].

Previous efforts to model global plastic waste emissions and movement through the environment have demonstrated the scale of the issue, highlighting large macroplastic emissions from countries with extensive coastlines, large populations and insufficient waste management[3–11]. Yet there is a growing understanding that a much higher (sub-national) resolution is required, which identifies plastic pollution hotspots and accounts for specific local solid waste management, behavioural, cultural and socio-economic conditions[12,17]. We believe that the very concept of 'emissions' also requires clarification, owing to the complexity of the phenomena (Methods and Extended Data Fig. 1). We use it here for clarity rather than the loosely defined terms of 'leakage' and 'mismanaged waste' described elsewhere[22] and we deliberately avoid the term 'release' suggested by the United Nations Economic Commission for Europe (UNECE)[23], which could imply deliberate activity. We define plastic emissions as material that has moved from the managed or mismanaged systems (in which waste is subject to a form of control, however basic; contained state) to the unmanaged system (the environment; uncontained state) with no control. We further classify emissions according to two categories: (1) debris (physical particles >5 mm) and (2) open burning (mass combusted in open uncontrolled fires). For clarification, open burning emissions relate to the mass of material that is subjected to the practice, rather than the gaseous, liquid or solid matter emitted by the process. Further definitions and scope are in Supplementary Information Section S.2.

Mapping and quantification of plastic waste material flows is hindered by the lack of sufficiently detailed and up-to-date records of waste management practices and quantities at a local level[24], which prevents the complete assessment of emissions from human systems[25]. Although coordinated work is underway to remedy this data paucity[24], a measurable baseline is urgently required to inform Plastics Treaty obligations[17]. As with greenhouse gas[18] or mercury[19,20] emissions inventories, this baseline would enable a more rational distribution of overseas development assistance, empower policymakers with scarce resources to develop evidence-based specialized national and sub-national strategies, action plans and targets[25], and create a strong evidential basis for the reorganization of material systems that have been the focus of Plastics Treaty proposals[26] and negotiations[27]. Therefore, we created a macroplastic emissions inventory using a new methodology to quantify emissions for 50,702 municipality-level administrations

[1]School of Civil Engineering, University of Leeds, Leeds, United Kingdom. ✉e-mail: C.Velis@leeds.ac.uk

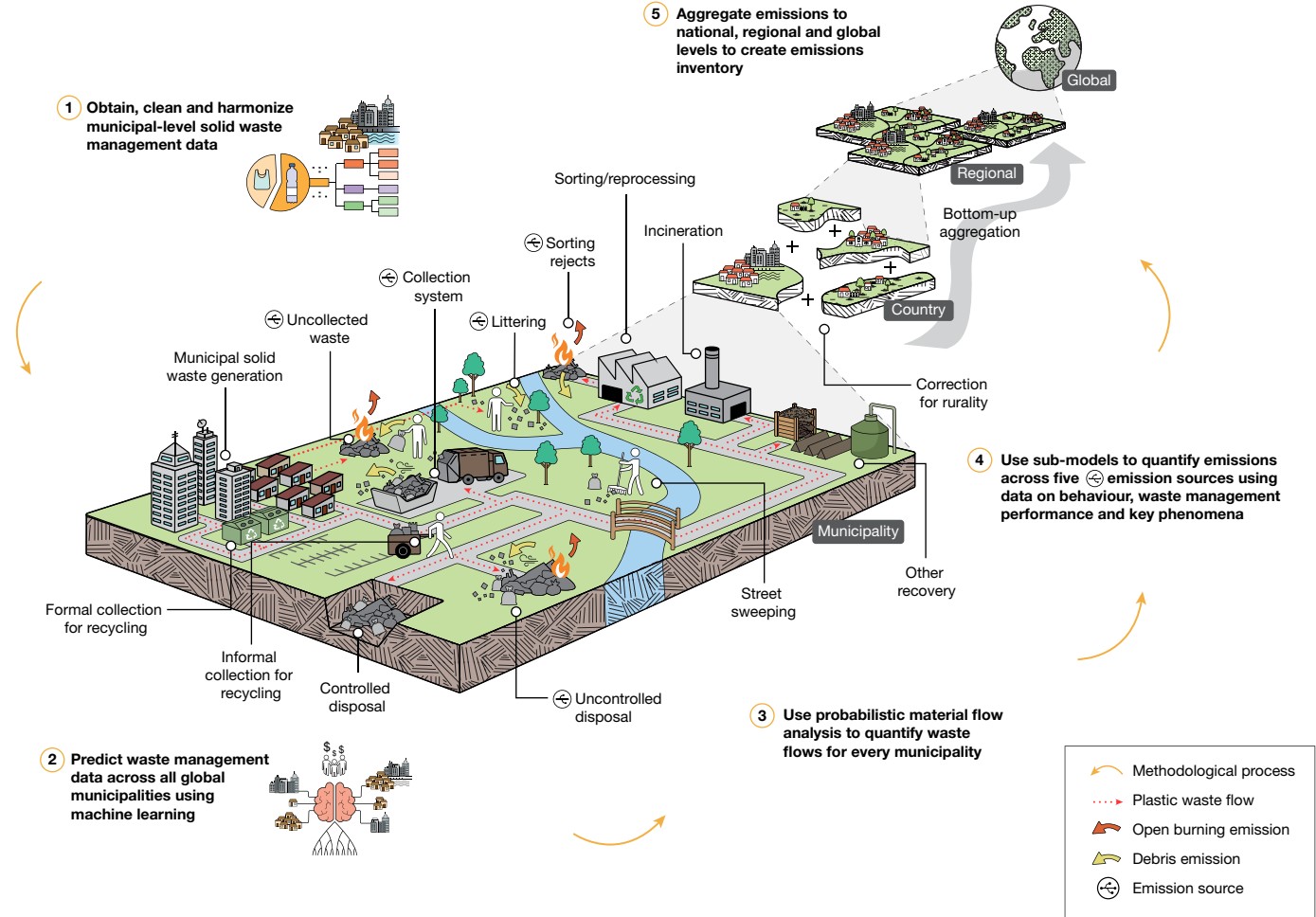

**Fig. 1 | Methodological process flow for creation of a global plastic pollution emissions inventory, as part of the 'Spatio-temporal quantification of plastic pollution origins and transport' model (SPOT).** Key plastic pollution sources and generalized waste management and circular economy flows are shown in this explanatory framework. Detailed materials and methods are available in the Supplementary Information.

from five land-based sources: (1) uncollected waste; (2) littering; (3) collection system; (4) uncontrolled disposal; and (5) rejects from sorting and reprocessing (Fig. 1). Unmeasured data were predicted using machine learning and flows were mapped using probabilistic material flow analysis (MFA) for the year 2020. See Methods and Supplementary Information for detailed methodology.

## Global emissions of plastic waste

We estimate that 52.1 Mt year$^{-1}$ [48.3–56.3] of macroplastic waste were emitted into the unmanaged system in 2020, representing 21% (wt.) of all the municipal plastic waste generated (251.7 Mt year$^{-1}$ [233.1–272.4]) globally (statistics reported are the arithmetic mean of all iterations—simulation runs; the 5th and 95th percentiles are in square brackets). Approximately 43% (wt.) (22.2 Mt year$^{-1}$ [20.6–24.0]) is unburned 'debris', meaning that it is no longer subject to any form of management or direct control and is at risk of transport across land and into the aquatic environment.

Most plastic pollution models do not report emissions in a way that is comparable with the present work, instead reporting emissions to 'the aquatic environment'[3], 'aquatic ecosystems'[6], 'the ocean'[8,28], 'mismanaged plastic waste'[5] and 'riverine outflows'[29]. However, two studies report comparable data. Ryberg et al.[11] estimated macroplastic debris emissions to the environment at 6.2 Mt year$^{-1}$ (confidence interval (CI): 2.0–20.4) in 2015. The upper end of the CI is within the range of our 5th percentile for debris emissions but the central estimate is

approximately 3.5 times lower than our mean. The categories reported by Ryberg et al.[11] include sea-based, industrial and construction sources, which are all outside the scope of our model. Removing these would reduce their central estimate to 4.9 Mt year$^{-1}$, 4.5 times lower than our mean estimate. The sum of 'terrestrial' and 'aquatic' emissions estimated by Lau et al.[9] for 2016 was 29 Mt (95% CI: 22–39). This estimate includes microplastics and material emitted at sea but is otherwise congruent with our debris emissions category. Although the average reported by Lau et al.[9] is approximately 23% higher than our mean estimate, the lower CI is approximately the same as our mean debris emissions.

Our model improves on earlier works and provides new information in five ways: (1) in this model, we used a bottom-up approach rather than regional[10] and archetypal[9] averages distributed to finer resolution (top-down approach); (2) our finer resolution accounts for spatial heterogeneity in sub-national waste management data; (3) we modelled emissions from five separate downstream sources rather than the single homogenous source used in other models[3–8,28]—'mismanaged (plastic) waste'[22], an umbrella term that encompasses a range of insufficiencies in waste management[12]; (4) our definition of 'emission' includes waste that escapes from 'dumpsites'[24] (defined in Methods) but excludes that retained within them because it is mostly buried beneath the waste mass[30] and poses a low risk of being blown or washed into the unmanaged system[31]. Only the 'working face' of these sites contains material at risk of transmission through the action of wind and surface water runoff[32] (Supplementary Information Section S.8.9). Conversely, it is

self-evident that waste that is uncollected, scattered on land or accumulated in smaller 'informal dumps' has a much higher probability of being mobilized and transported across the terrestrial surface and into the aquatic environment; and 5) We account for the open burning of waste (Supplementary Information Section S.8.11), which is not specifically considered in most plastic pollution models[3–8,11,28] and which our results indicate contributes to 57% (29.9 Mt year[-1] [27.6–32.4]) of all plastic waste emitted, resulting in widespread risk to human health and the environment[33]. As far as we are aware, only Lau et al.[9] report a comparable estimate of open burning of municipal solid waste plastic of 49 Mt year[-1] (95% CI: 40–60) for 2016, two-thirds more than our estimate. The reason for this difference is the method of calculation. Whereas Lau et al.[9] used emission factors derived from expert assumptions published by the Intergovernmental Panel on Climate Change (IPCC)[18] and extrapolated from Wiedinmyer et al.[34], our study uses census and survey activity data from 44 countries (Supplementary Information Section S.8.11).

## Plastic emission hotspots outlook

On an absolute basis, we find that plastic pollution emissions are highest across countries in Southern Asia, Sub-Saharan Africa and South-eastern Asia (Fig. 2a–c), with the largest amount (9.3 Mt year[-1] [6.5–12.7]) emitted by India, equivalent to nearly one-fifth of global plastic emissions. In contrast to previous plastic pollution models that positioned China as the world's highest plastic polluter[5,8], it is ranked fourth in our results, with emissions of 2.8 Mt year[-1] [2.1–3.7], less than Nigeria (3.5 Mt year[-1] [2.6–4.6]) and Indonesia (3.4 Mt year[-1] [2.5–4.3]). This lower contribution to plastic emissions from China reflects our use of more up-to-date data[35] that shows its substantial progress in adopting waste incineration and controlled landfill[36]. Conversely, India reports that its dumpsites (uncontrolled land disposal) outnumber sanitary landfills by 10:1 (ref. 37) and, despite the claim that there is a national collection coverage of 95%, there is evidence that official statistics do not include rural areas, open burning of uncollected waste or waste recycled by the informal sector[38]. This means that India's official waste generation rate (approximately 0.12 kilograms per capita per day (kg cap[-1] day[-1])) is probably underestimated and waste collection overestimated. Our model corrects for flows missing in officially reported statistics, resulting in a waste generation rate for India of 0.54 kg cap[-1] day[-1] [0.39–0.73], which is similar to and between other comparable estimates[38–40].

Our data for India indicate a collection coverage of 81% [80–82], meaning that nearly 53% (wt.) [51–56] of the country's plastic waste emissions (30% wt. [29–32] debris and 23% wt. [22–25] open burning) come from the 255 [241–270] million people (18% [17–19] of the population) whose waste is uncollected. Most of the remaining emissions (38% wt. [36–40]) are as a result of open burning on dumpsites, in which fires are reported to be common[38]. Overall, we estimate that 56.8 Mt year[-1] [40.0–77.7] of municipal solid waste is open burned in India, of which 5.8 Mt year[-1] [4.1–7.9] is plastic. This is within the lower end of the ranges modelled by Chaudhary et al.[38] of 74.0 Mt year[-1] (uncertainty: 30–92) and Sharma et al.[39] of 68 Mt year[-1] (range: 45–105).

Open burning rather than intact items (debris) is the predominant emission type across most United Nations sub-regions, except for those which are predominantly in the Global North (Northern America, Northern Europe, Western Europe and Australia and New Zealand) and Sub-Saharan Africa, in which debris emissions (7.4 Mt year[-1] [6.7–8.2]) are slightly higher than open burning emissions (5.9 Mt year[-1] [5.2–6.6]) (Fig. 2c). This result is driven by data that indicate lower levels of open burning in the rural areas of low-income countries (LICs), of which there are many in the Sub-Saharan Africa region (Supplementary Fig. S.24d,f).

Approximately 69% (35.7 Mt year[-1]) of the world's plastic waste emissions come from 20 countries, of which four are LICs, nine are lower middle-income countries (LMCs) and seven are upper middle-income countries (UMCs). Despite high-income countries (HICs) having higher plastic waste generation rates (0.17 kg cap[-1] day[-1] [0.15–0.20]), none are ranked in the top 90 polluters, because most have 100% collection coverage and controlled disposal. Furthermore, our modelling accounts for the mitigating impact of street sweeping activity on emissions, which is greater in HICs (Supplementary Information Section S.8.5). We acknowledge that we may have underestimated plastic waste emissions from some HICs because we deliberately excluded plastic waste exports from our analysis. As explained in Supplementary Information Section S.2, plastic waste exports from the top ten Organisation for Economic Co-operation and Development (OECD) exporters to non-OECD countries and Turkey have substantially decreased from nearly 5.4 Mt year[-1] in 2017 to less than 1.7 Mt year[-1] in 2022 (ref. 41), contributing approximately 0.03 Mt year[-1] of emissions. Although this might affect some individual country results, the overall effect would be negligible in comparison with other sources.

Countries in low-income and middle-income categories have much lower plastic waste generation (LICs: 0.04 kg cap[-1] day[-1]; LMCs: 0.07 kg cap[-1] day[-1]; UMCs: 0.10 kg cap[-1] day[-1]). However, in contrast to HICs, a large proportion of it is either uncollected (LICs: 55% wt.; LMCs: 26% wt.; UMCs: 11% wt.) or disposed of in dumpsites (uncontrolled disposal) (LICs: 36% wt.; LMCs: 57% wt.; UMCs: 19% wt.). The nine countries that make up the Southern Asia region emit a similar amount of plastic waste (15.1 Mt year[-1] [12.1–18.7]) to the 51 countries in Sub-Saharan Africa (13.3 Mt year[-1] [12.0–14.7]) (Fig. 2b,c), with Nigeria contributing to approximately one-quarter (3.5 Mt year[-1] [2.7–4.6]) of the Sub-Saharan African burden. Urban areas (cities, towns and semi-densely populated areas) account for most emissions in all regions (Fig. 2b) because of low rural populations (Supplementary Information Section 7.1) and much lower plastic waste generation. However, we acknowledge that notable data gaps on solid waste management in rural communities exist and future efforts to address plastic pollution must include these often overlooked communities[42].

Flexible plastic debris has a higher probability of being emitted into the environment in the Global South compared with rigid debris (mean ratio 56:44), driven by its greater prevalence (waste composition) and its propensity for mobilization under the action of wind and surface water (Fig. 2d). In the Global North (for example, Northern America), the opposite is true (mean ratio 33:67) because rigid plastics are more prevalent in the waste and because emissions are driven by littering rather than meteorological forcing.

## Per-capita emission hotspots

The contrast between absolute plastic waste emissions from the Global North and the Global South is stark (Fig. 3a,c). However, on a per-capita basis, insufficiencies in local and national waste management systems are more apparent (Extended Data Figs. 2–6). For example, China, the world's fourth largest absolute emitter, is one of the least polluting UMCs, ranked 153 of all countries on a per-capita basis (1.97 kg cap[-1] year[-1] [1.48–2.61]), and India, the world's largest absolute emitter, is ranked 127 on a per-capita basis (6.64 kg cap[-1] year[-1] [4.66–9.08]). Conversely, Russia, the world's fifth largest emitter on an absolute basis, also has high emissions on a per-capita basis (11.71 kg cap[-1] year[-1] [7.80–16.17]) because it is reported to have very low levels of controlled disposal[43,44]. Many countries in Sub-Saharan Africa that show low absolute plastic emissions are hotspots on a per-capita basis (Extended Data Fig. 4). Given the anticipated population boom in the region[45], it is conceivable that, with an average emission rate of 12.01 kg cap[-1] year[-1] [10.83–13.25], Sub-Saharan Africa will become the world's largest absolute source of plastic pollution within the next few decades.

Municipal-scale probability distributions indicate substantial uncertainty within municipalities for some of our model outputs (Fig. 3b).

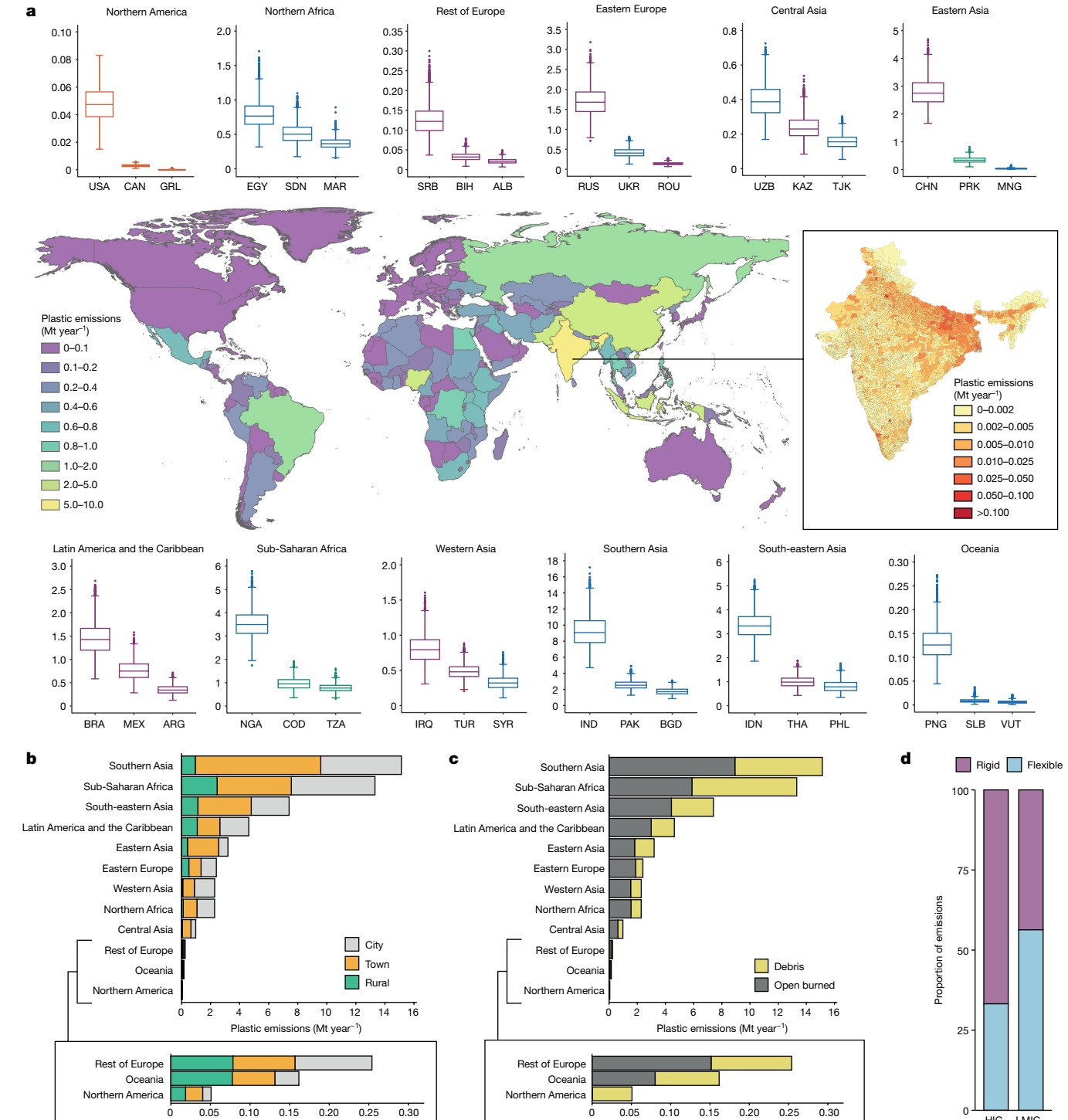

**Fig. 2 | Macroplastic emissions into the environment (debris and open burned plastic) in Mt year⁻¹ for the year 2020. a**, Mean macroplastic emissions by country. Inset illustrates mean municipal-level emissions for India, from which the national results are calculated. Box plots show distribution of probabilistic material flow analysis results for the three highest macroplastic emitting countries in each United Nations sub-region. Box plot statistics: lower and upper hinges correspond to the first and third quartiles and the central line is the median. Whiskers extend to the data point no further than 1.5 times the interquartile range from the hinge, with outlier values beyond this denoted as dots. **b**, Emissions by United Nations sub-region and settlement typology[54]. Two groups of United Nations sub-regions are merged for simplicity into 'Rest of Europe' (Northern Europe, Southern Europe, Western Europe) and 'Oceania' (Polynesia, Australia and New Zealand, Melanesia, Micronesia). **c**, Mean emissions by United Nations sub-region and emission type. **d**, Mean proportion of macroplastic emissions by plastic format for the income categories of HIC and low-income or middle-income countries (LMIC).

For example, the 5th and 95th percentiles of plastic emissions are 0.77–11.87 kg cap⁻¹ year⁻¹ (median 3.62 kg cap⁻¹ year⁻¹) for Agra (India) and 0.11–4.72 kg cap⁻¹ year⁻¹ (median 0.34 kg cap⁻¹ year⁻¹) for Maracaibo (Venezuela). The large ranges within many municipalities and relatively high kurtosis, for example, Shenzhen (42.3) and Maracaibo (19.9), are a consequence of our conservative application of probability density functions for many of the model's input data, which have propagated through to the results.

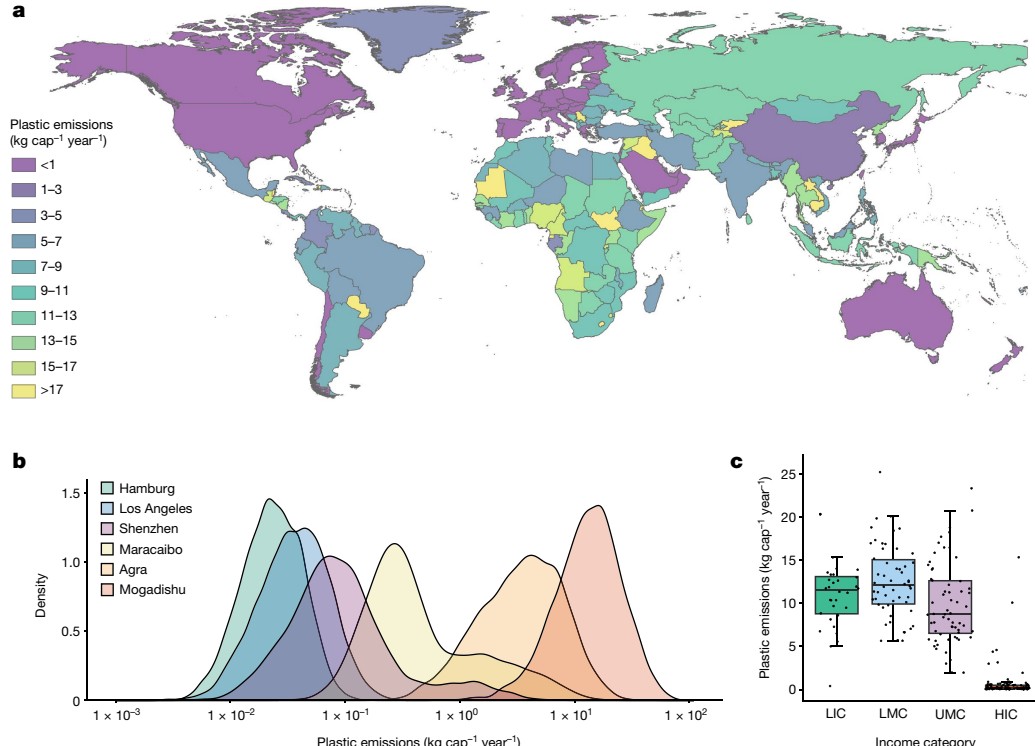

**Fig. 3 | Macroplastic emissions into the environment (debris and open burned) in kg cap⁻¹ year⁻¹ for the year 2020. a**, Mean macroplastic emissions by country. **b**, Probability distributions of macroplastic emissions for six global cities >1 million population. **c**, Country-level macroplastic emissions by income category. Black dots are individual country results in each income category. The lower and upper hinges of the box plots correspond to the first and third quartiles and the central line is the median. Whiskers extend to the data point no further than 1.5 times the interquartile range from the hinge.

Despite the wide uncertainty within each municipality, there are very large differences between many of them, enough to differentiate the most challenging locations from the least (Fig. 3b). For example, median plastic emissions for Hamburg (Germany) are estimated at 0.02 kg cap⁻¹ year⁻¹ [0.01–0.06] compared with Mogadishu (Somalia), which has almost 680 times more (13.63 kg cap⁻¹ year⁻¹ [4.05–36.70]). Such large differences demonstrate that substantial reductions in plastic emissions are feasible, reiterating the importance of measuring sound solid waste management activity data. Continuing efforts to gather more reliable municipal-scale information[24] for SDG indicator 11.6.1 will gradually improve the accuracy of our model. However, much more comprehensive measurement and monitoring is required to improve the accuracy of flows that are rarely measured and that have been populated here using our conceptual sub-models.

## Sources of plastic emissions

Uncollected waste is the largest contributor to plastic pollution in the Global South, accounting for 68% (35.6 Mt year⁻¹) of all plastic waste emissions and 85% (18.7 Mt year⁻¹) of all debris emissions. On a per-capita basis, uncollected waste represents 69%, 66% and 80% (wt.) of emissions in UMCs, LMCs and LICs, respectively (Fig. 4b). Approximately 56% (19.9 Mt year⁻¹ [17.8–22.3]) of emissions from uncollected waste come from LMCs, in which the mean collection coverage is 74% [72–75] (Fig. 4a). Uncollected waste in LMCs accounts for 38% of total global plastic emissions and 51% (11.3 Mt year⁻¹) of debris emissions. As far as we are aware, none of the other global plastic pollution models[3–8,11,28] has explicitly highlighted uncollected waste as the main source of plastic pollution, instead grouping it in the 'mismanaged waste' category or, in one case[9], together with disposal site debris emissions. Here we show that plastic waste emissions from uncontrolled land disposal sites (dumpsites), although important, contribute 25% (12.8 Mt year⁻¹ [11.5–14.3]) of global plastic waste emissions, of which 98% (wt.) is open burned. This means that just 0.25 Mt year⁻¹ is emitted from land disposal sites as debris, approximately 0.4% (wt.) of plastics deposited in uncontrolled disposal sites worldwide. This is substantially less than has been inferred elsewhere. For example, Lau et al.[9] estimated that between 1% and 1.5% of rigid plastics and 8% and 13% of flexible and multimaterial plastics deposited in uncontrolled disposal sites would reach the aquatic environment each year. The difference is that Lau et al.[9] used expert judgement to derive their transfer coefficients, whereas this work used a more detailed sub-model based on the surface area and runoff characteristics of dumpsites detailed in Supplementary Information Section S.8.9.

HICs contribute 0.3% (0.16 Mt year⁻¹ [0.14–0.19]) of global plastic waste emissions. Among HICs, uncollected waste is the source of 21% [15–27] (0.03 Mt year⁻¹ [0.02–0.05]) of plastic waste emissions, just 0.06% of the global emissions burden, largely because collection coverage is nearly 100%. The largest source of debris emissions in HICs is littering (see 'Uncollected litter' defined in Supplementary Table S.3), accounting for 53% of debris emissions and 49% (0.08 Mt year⁻¹, 0.06 kg cap year⁻¹) of all plastic emissions in the Global North (Fig. 4a,b). Of this, 0.03 Mt year⁻¹ takes place in Northern America and 0.03 Mt year⁻¹ in the Rest of Europe region (Fig. 4c), representing 0.09 kg cap year⁻¹ and 0.07 kg cap year⁻¹, respectively (Fig. 4d). The behavioural nature of littering[46] contrasts with the underlying drivers of other emission sources, especially those in the Global South. This is because, although littering is negatively correlated with waste receptacle provision[47], it is largely driven by the decisions of individuals[46]. By contrast, the 1.5 billion individuals whose waste is uncollected in the Global South have little choice but to self-manage it (defined in Supplementary Information Section S.4.1).

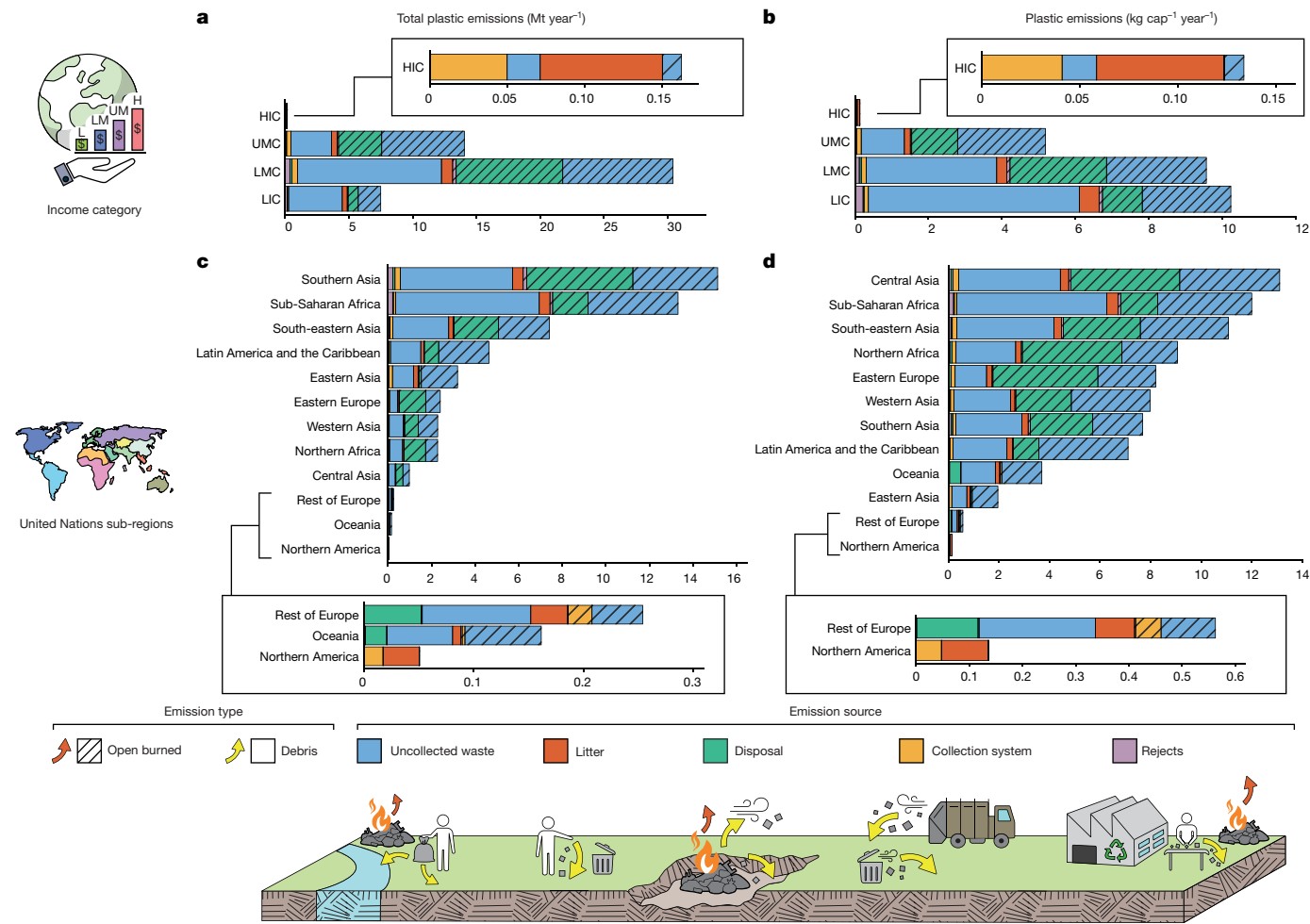

**Fig. 4 | Mean macroplastic emissions into the environment by emission source and emission type (open burning and debris) for the year 2020.** Shown by: **a**, absolute mass and income category; **b**, per capita and income category; **c**, absolute mass and United Nations sub-region; and **d**, per capita and United Nations sub-region. Absolute mass of emissions (**a,c**) has unit Mt year⁻¹, whereas per-capita emissions (**b,d**) has unit kg cap⁻¹ year⁻¹. Two groups of United Nations sub-regions are merged for simplicity into 'Rest of Europe' (Northern Europe, Southern Europe, Western Europe) and 'Oceania' (Polynesia, Australia and New Zealand, Melanesia, Micronesia). LIC, low-income country; LMC, lower middle income country; UMC, upper middle income country; HIC, high-income country.

The mismanagement of rejects from plastics sorting and reprocessing (recycling system) in both the Global North and the Global South results in 1.0 Mt year⁻¹ [0.9–1.1] of plastic waste emissions to the environment. These emissions have often been the focus of attention, particularly in relation to the transboundary trade (exports) in waste plastics[48]. However, here we show that the emissions burden from recycling macroplastic rejects is comparatively very small.

## An inventory to support the treaty

The purpose of our study was to create a macroplastic pollution inventory method for baselining and monitoring emissions at the local scale, at which on-the-ground actions can be applied. Such an emissions inventory, explaining the mechanisms for emission from the waste management and societal systems, could form a basis for a more detailed and comprehensive assessment of possible interventions. Once macroplastics have entered the environment, they are technically and economically challenging to remove[49] and, over time, will inevitably fragment into innumerable microplastics[50], making clean-up efforts even more challenging. Minimizing plastic pollution at source by preventing the emission event in the first place must be a priority of the Plastics Treaty[17] and our insight indicates that tackling uncollected waste would have a bigger impact than mitigating all other land-based macroplastic sources combined. Notably, we already have a large global workforce of informal recyclers, entrepreneurs who our model shows collect more than 49.8 Mt year⁻¹ [45.1– 54.9] of waste plastics annually, much of which would otherwise be mismanaged.

We suggest that interventions to reduce uncollected plastic waste would focus on upstream material reduction to reduce waste generation and/or substantial improvement of waste collection services and infrastructure, and our emissions inventory sets a detailed basis for this. As highlighted elsewhere[9,51], mitigating plastic waste emissions will require a multisectoral approach that includes addressing insufficiencies across the lifecycle, including redesign of product systems, source reduction and improving recycling systems worldwide. The plausibility of timely and at-scale deployment of such interventions needs to be carefully reassessed in the context of our new results.

The large mass of waste that is burned in open uncontrolled fires has not formed a central part of discussions at Plastics Treaty negotiations[26,27]. Yet, according to our model, more plastic waste is burned than is emitted as debris worldwide, releasing a cocktail of potentially hazardous substances and climate forcing emissions, which may have

a substantial impact on human health and ecological systems[33]. An unintended consequence of interventions to mitigate the release of debris could result in an increase in emissions from open burning and vice versa[52]. Therefore, we propose that the inclusion of this phenomenon is a critical component of the forthcoming negotiations: clearly, choosing between two main forms of plastic pollution should not be an option.

We acknowledge that there is a dearth of robust, quality-controlled and verifiable waste management activity data. We have tediously screened, assessed, harmonized and corrected relevant data, incorporating uncertainty using a probabilistic approach. We have designed a conceptual framework that allows the model's input data and structure to be continuously updated. As more quality-controlled locally obtained measurements from across the waste and resources system become available, and our understanding of release mechanisms improves, the model's precision and accuracy can be ameliorated.

As with international climate change agreements[53], signatories to the Plastics Treaty will require a method to calculate and baseline their plastic waste emissions so that they can compare them with others. Our emissions inventory enables them to carry out these estimates at high resolution by conceptualizing the mechanisms of emission, providing insights into the nature, extent and causes of plastic pollution and, therefore, enabling development of evidence-based national and sub-national action plans to eliminate plastic in our environment.

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

# Methods

We created a macroplastic emissions inventory using a new methodology to quantify emissions from land-based sources for 50,702 municipality-level administrations[55] (see Supplementary Information for details on the method). We define plastic emissions as material that has moved from the managed or mismanaged systems (in which waste is subject to a form of control, however basic) to the unmanaged system (the environment) with no control. For example, open dumpsites, defined here as structures that contain concentrations of collected waste with only basic control to prevent its interaction with the environment, are a form of control, because most of the material buried beneath the waste mass is unlikely to undergo further movement into the environment.

Material was mapped through 81 downstream (after-use phase) processes to simulate the flow of municipal solid waste through globally diverse waste management systems (Fig. 1 and Supplementary Information Section 4). Emissions of land-based macroplastic debris (physical particles >5 mm) and open burning (combustion in open uncontrolled fires) from municipal solid waste (defined in Supplementary Information Section S.2) were quantified for flexible and rigid plastics (format). Activity data (the intensity of waste and resources recovery management activity) were obtained from four global[56–59] and two national[35,60] waste management databases. These were checked for errors, harmonized to a consistent basis and corrected if necessary, creating the first comprehensively quality controlled city-level solid waste management database with worldwide coverage (Supplementary Data 1). Our primary input data represent 12.2% of the 2015 global population, spanning each of the World Bank income categories (LICs: 12.0%; LMCs: 11.4%; UMCs: 13.5%; HICs: 11.2%). Further discussion on the representativeness of our input data is presented in Supplementary Information Section S.6.7.

Quantile regression random forest models[61] predicted data for all global municipalities using national and sub-national socio-economic indicators. Waste management, circular economy and plastic waste emission characteristics, variables that are not commonly measured or reported, were estimated using data from the literature or through the creation of new conceptual models. These newly developed 'sub-models' (Supplementary Information Sections S.8.2, S.8.3, S.8.3.4, S.8.5, S.8.5.2, S.8.8, S.8.9, S.8.11.1 and S.9.1.2) used data on human behaviour, material value, socio-economic development, population density and solid waste management performance, creating an explanatory framework through which to estimate unmeasured system characteristics. The use of 'process-level sub-models' to describe larger systems has recently been advocated for plastic pollution modelling[13].

Probabilistic (Monte Carlo simulation) MFA mapped flows of municipal solid waste (5,000 iterations) throughout the system (Supplementary Information Section S.4), resulting in detailed information on municipal solid waste and plastic waste management for each global municipality (Supplementary Data 5). Emissions into the unmanaged system, defined here as uncontained waste that is no longer subject to any form of management or control, were estimated for five key sources: (1) uncollected waste; (2) littering; (3) collection system; (4) uncontrolled disposal; and (5) rejects from sorting and reprocessing (Extended Data Fig. 1). The probabilistic MFA used probability density functions from two sources: (1) the results of the machine learning predictions and (2) from the secondary data collection and processing step detailed in Supplementary Information Section S.8. A full list of probability density functions used in our model is available in Supplementary Data 6 and the MFA equations are shown in Supplementary Data 2.

These flows and their associated uncertainty were aggregated to the national scale (Supplementary Data 3) to align with reporting for SDG indicator 11.6.1 (ref. 24) and to the regional and global scales (Supplementary Data 4) to create a multiresolution global plastic emissions inventory (Fig. 1 and Extended Data Fig. 7). This inventory is the first-stage prerequisite for a second terrestrial transport model (not discussed further here), collectively named the 'Spatio-temporal quantification of plastic pollution origins and transport' model (SPOT). Although we acknowledge that upstream processes during the production, conversion and use phases result in a range of emissions from plastics, they are outside the scope of our modelling. We also exclude textiles, sea-based sources of plastic pollution and waste electrical and electronic equipment. To improve comprehension of proportionality, the results are reported as the mean of all iterations (simulation runs). Numbers in square brackets are the 5th and 95th percentiles of all iterations. As there are no datasets with which to validate our model outputs, we took the same approach as Lau et al.[9] and carried out global sensitivity analysis to assess the influence of the model inputs and structure on its results (Supplementary Information Section S.10).

We warn readers to consider the full uncertainty in our MFA results, particularly for municipal-scale outputs at which the ranges are generally much larger than national-scale or regional-scale aggregations. The origins of uncertainty in our model are discussed at length in Supplementary Information Section S.9.2.2. We also explain in Supplementary Information Section S.9.1.1 a specific circumstance in which we decided not to quantify uncertainty for the uncontrolled disposal coefficient (tC3) owing to limitations of the quantile regression random forest predictive capability for that particular aspect of the system.

## Data availability

Supplementary Data 1–6 are freely available as part of the Supplementary Information and are available from Dryad: https://doi.org/10.5061/dryad.8cz8w9gxb. Administrative boundaries used for the maps were sourced from GADM version 3.6 and are available from https://gadm.org/.

## Code availability

All code, model inputs and outputs are available from Dryad: https://doi.org/10.5061/dryad.8cz8w9gxb.

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

**Acknowledgements** We would like to thank Y. Gavish, data analyst, for comments on modelling uncertainty and machine learning; A. Savvantoglou for illustrations and graphic design; C. Gonzales and M. Harkness for data cleaning; K. Terzidis for collection and analysis of incineration data. For assistance in securing access to municipality-related primary data and assisting with comprehension of data collection and reporting methods used by the main international datasets: N. Takeuchi (UN-Habitat), A. Whiteman (Wasteaware), M. Newbury (United Nations Statistics Division) and S. Kaza (World Bank Group). Funding: this work was partly supported by the United Nations Human Settlements Programme (UN-Habitat), with further in-kind support by the University of Leeds. The views expressed in this article are those of the authors' alone and do not necessarily represent the views or policies of UN-Habitat.

**Author contributions** J.W.C.: conceptualization; methodology; software; validation; formal analysis; investigation; data curation; writing – original draft; writing – review and editing; visualization. E.C.: conceptualization; methodology; validation; formal analysis; investigation; data curation; writing – original draft; writing – review and editing; visualization. C.A.V.: conceptualization; methodology; validation; formal analysis; investigation; data curation; writing – original draft; writing – review and editing; visualization; supervision; resources; funding acquisition.

**Competing interests** C.A.V. consults for organizations active in the waste, resources and circular economy sphere. He receives funding from UKRI, GCRF, NERC, ESRC, BBSRC, Royal

# Article

Academy of Engineering, British Council, Innovate UK, EC H2020, World Bank Group, OECD, GIZ, UN-Habitat, UNESCAP, UNOPS, The Pew Charitable Trusts, IGES, ISWA, GRID-Arendal, Swedish EPA and SYSTEMIQ. He is affiliated with the International Solid Waste Association (ISWA), the Scientist's Coalition for an Effective Plastics Treaty and the Innovation Alliance for a Global Plastics Treaty. The University of Leeds has memorandums of understanding with the Alliance To End Plastic Waste and the United Nations Environment Global Partnership on Plastic Pollution and Marine Litter (GPML), which refer to plastic pollution databases. E.C. has consulted for Tearfund.

**Additional information**
**Correspondence and requests for materials** should be addressed to Costas A. Velis.

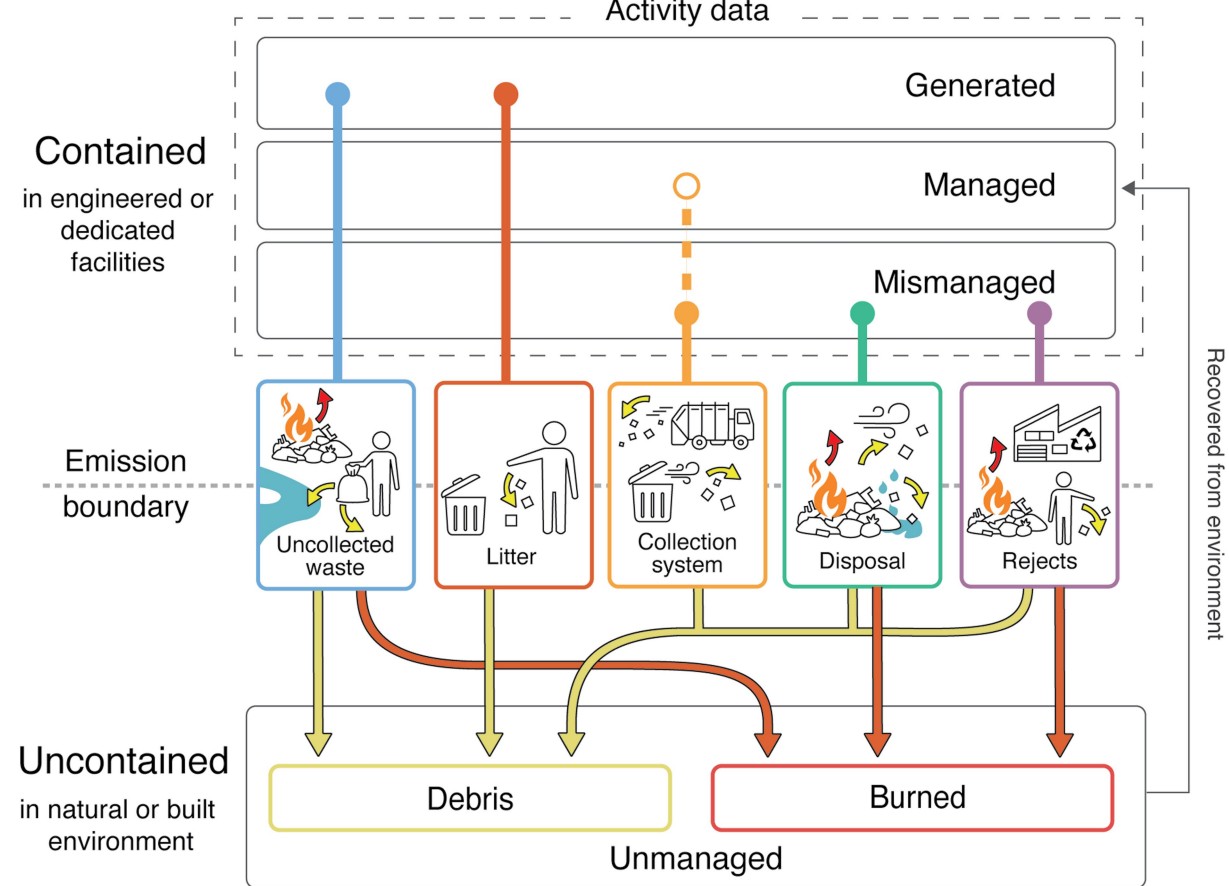

**Extended Data Fig. 1 | The point at which material passes from a contained to an uncontained state across the emission boundary is described here as an emission.** Emissions originate from five core emission sources and from three system parts (generated, managed and mismanaged), each of which exhibit different containment characteristics.

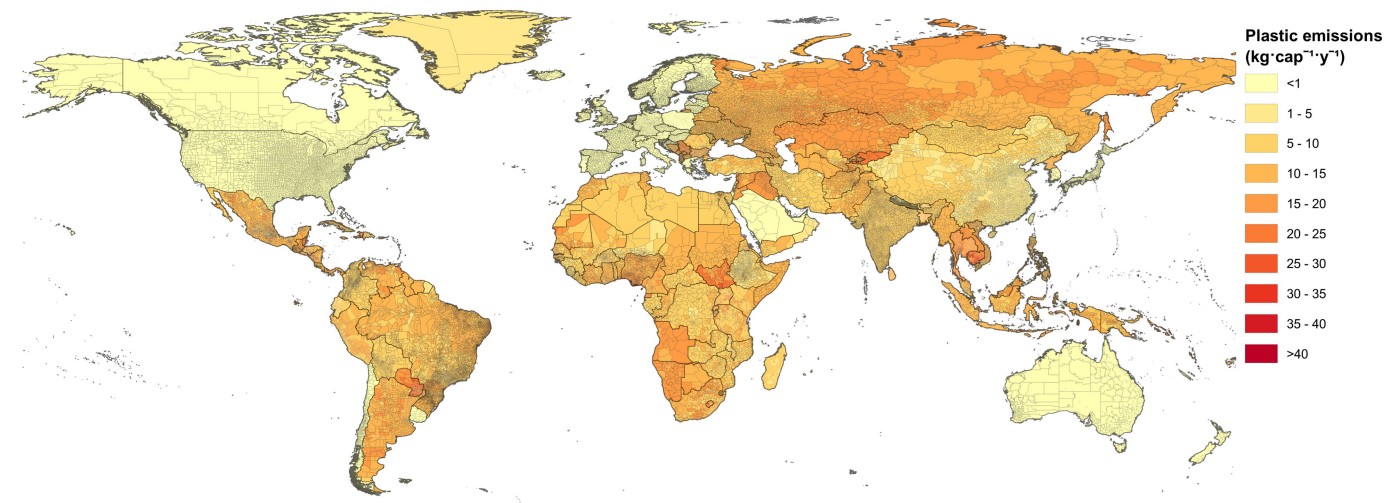

**Extended Data Fig. 2 | Macroplastic emissions into the environment (debris and open burned) by municipality in mean kg cap⁻¹ year⁻¹ for the year 2020.** Countries in the Global South have high per-capita emissions compared with those in the Global North.

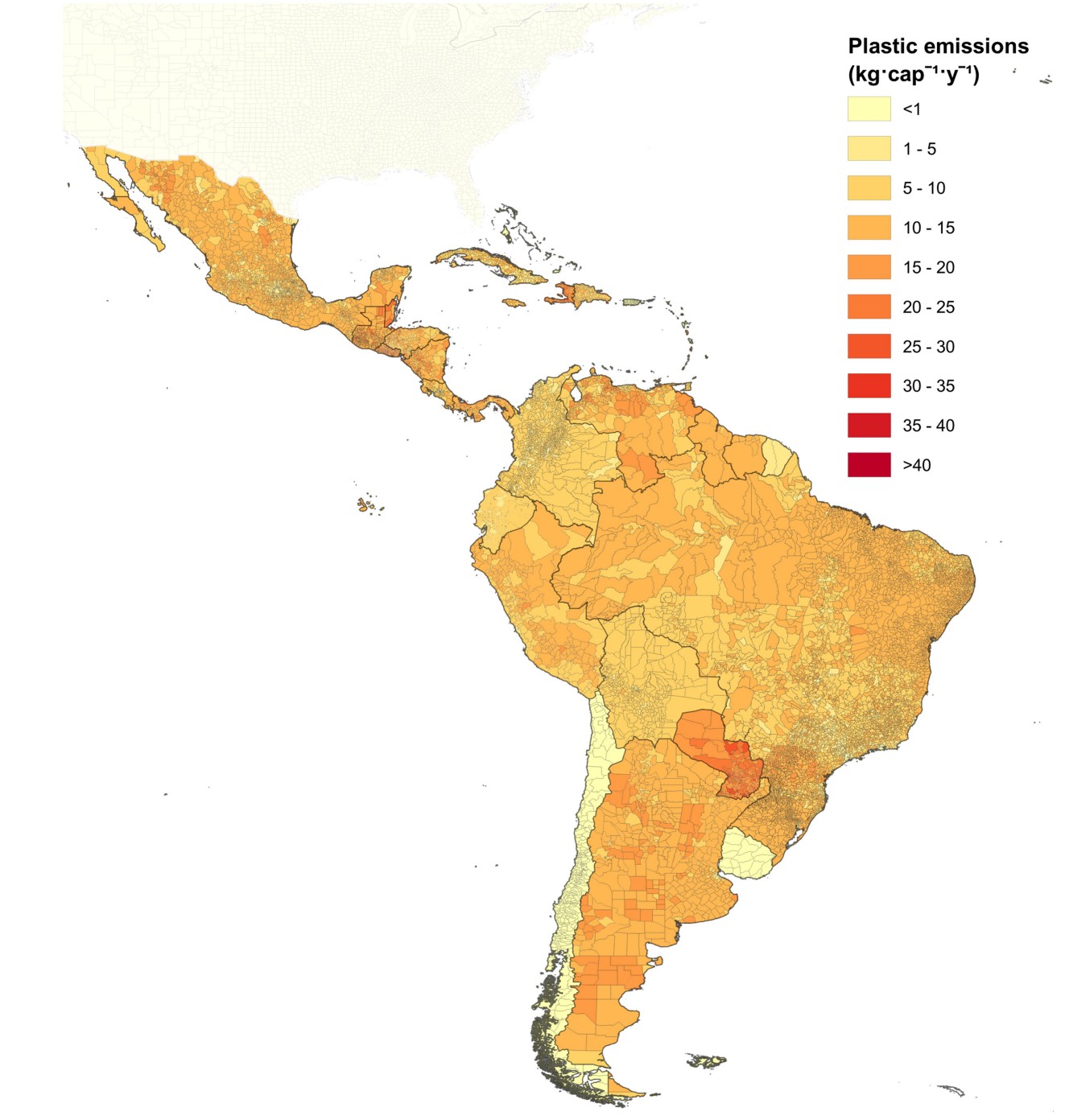

**Plastic emissions (kg·cap⁻¹·y⁻¹)**

- <1
- 1 - 5
- 5 - 10
- 10 - 15
- 15 - 20
- 20 - 25
- 25 - 30
- 30 - 35
- 35 - 40
- >40

**Extended Data Fig. 3 | Macroplastic emissions into the environment (debris and open burned) by municipality for Latin America and the Caribbean in mean kg cap⁻¹ year⁻¹ for the year 2020.** Hotspots for per-capita emissions include municipalities in Paraguay, Belize and Haiti, whereas municipalities in Uruguay and Chile have comparably lower emissions.

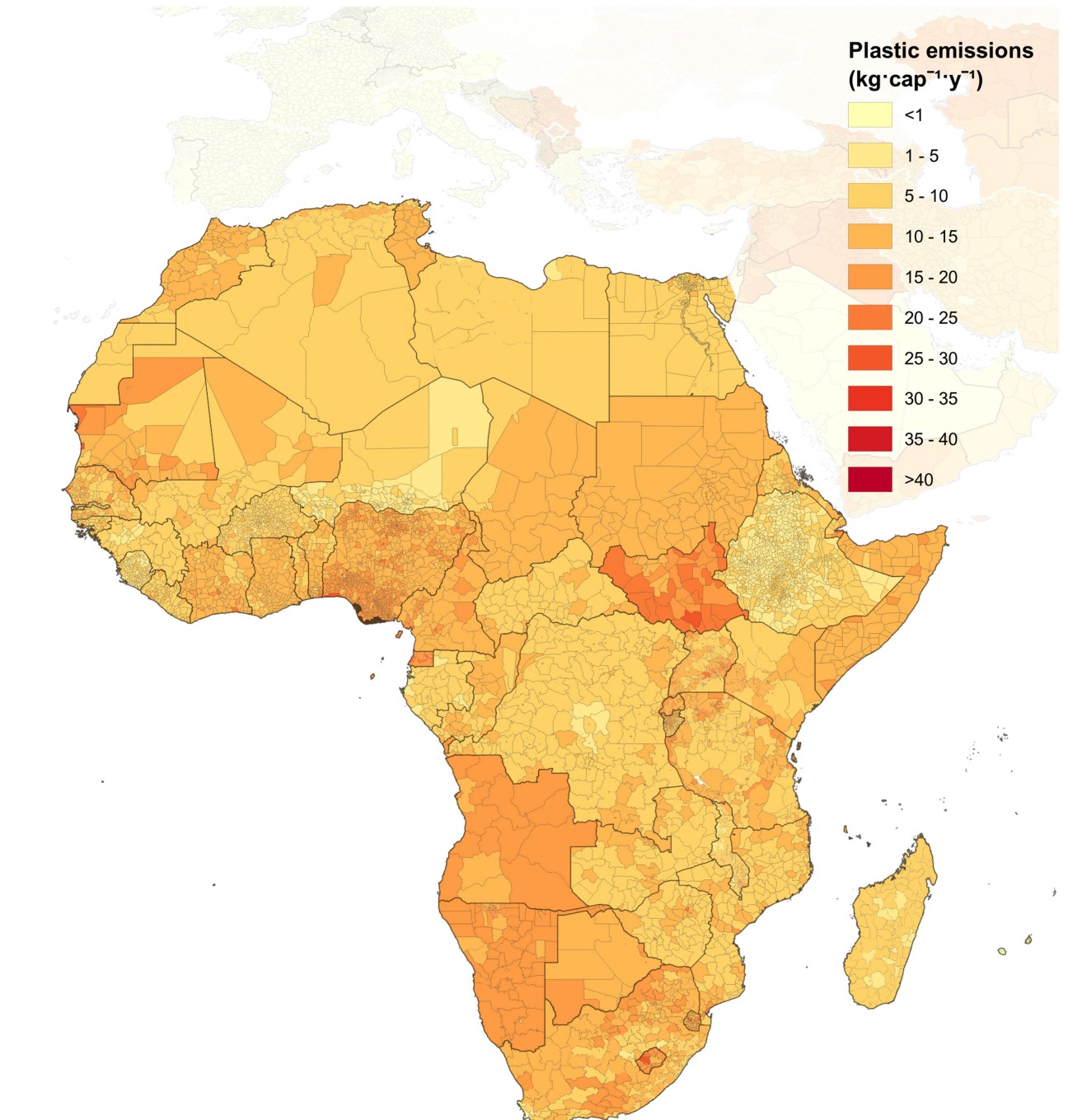

**Extended Data Fig. 4 | Macroplastic emissions into the environment (debris and open burned) by municipality for Africa in mean kg cap⁻¹ year⁻¹ for the year 2020.** Per-capita emissions are high throughout the continent, with notable hotspots including municipalities in South Sudan, Angola and Namibia. Several megacities stand out as key hotspots, including Lagos (Nigeria), Juba (South Sudan) and Nouakchott (Mauritania).

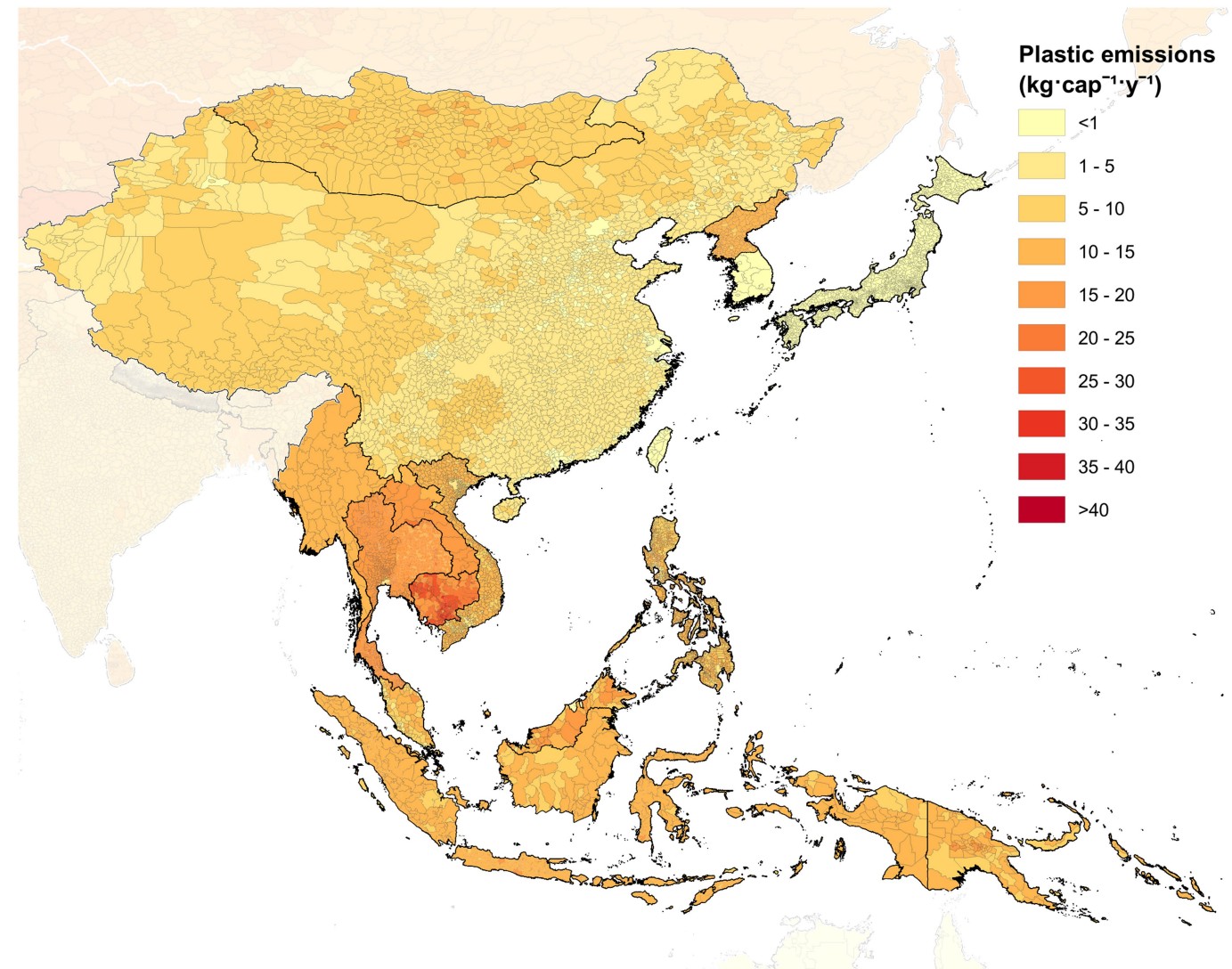

**Extended Data Fig. 5 | Macroplastic emissions into the environment (debris and open burned) by municipality for Eastern Asia and South-eastern Asia in mean kg cap⁻¹ year⁻¹ for the year 2020.** Emissions on a per-capita basis are low for municipalities in HICs, such as Japan and South Korea, but high throughout much of South-eastern Asia, particularly Cambodia. Eastern China has low per-capita emissions owing to recent progress in solid waste management. However, emissions are marginally higher in Western China.

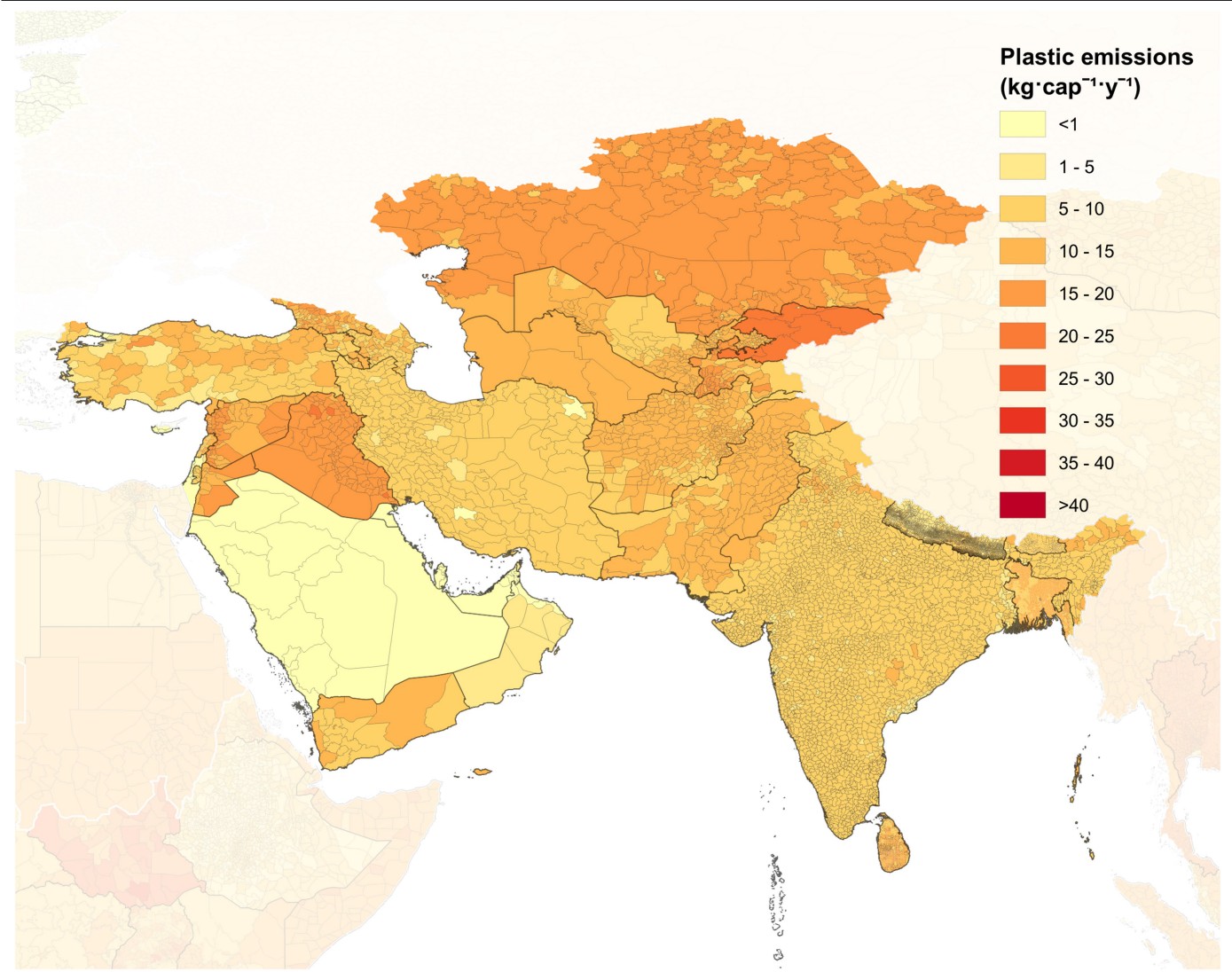

**Extended Data Fig. 6 | Macroplastic emissions into the environment (debris and open burned) by municipality for Central Asia, Western Asia and Southern Asia in mean kg cap⁻¹ year⁻¹ for the year 2020.** Per-capita emissions are high throughout the region, with the exception of municipalities in HICs on the Arabian Peninsula, such as Saudi Arabia, Qatar and United Arab Emirates.

Municipalities in Kyrgyzstan, Kazakhstan, Iraq, Jordan and Syria have relatively high per-capita emissions. Although India has the highest absolute emissions of all countries, on a per-capita basis, most of its municipalities have between 5 and 10 kg cap⁻¹ year⁻¹.

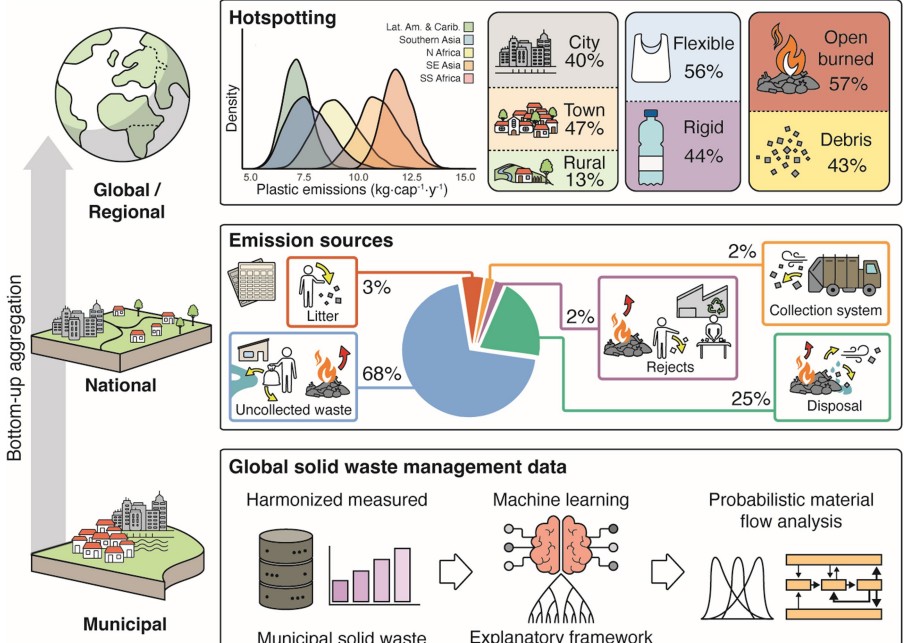

**Extended Data Fig. 7 | Graphical abstract for a local-to-global emissions inventory of macroplastic pollution.** Municipal level data were cleaned, harmonized and used to train a quantile regression random forest machine learning model, which was used to generate core material flow data for 50,702 municipalities worldwide. These data, combined with explanatory conceptual submodels, were used to populate and define flows in a probabilistic material flow analysis model (Monte Carlo) with 81 processes. The results are presented at municipal level, which showed a large variations in emissions, and then as aggregations at national, income category and global levels. The majority of emissions come from uncollected waste, whereas litter accounts for a comparatively small proportion worldwide. Of the 52.1 Mt year$^{-1}$ (mean) of emissions produced, approximately 57% wt. are burned in open uncontrolled fires.