## [Peer Review File · Nature]

Manuscript Title: A local-to-global emissions inventory of macroplastic pollution

Reviewer Comments & Author Rebuttals

Reviewer Reports on the Initial Version:

Referee #1 (Remarks to the Author):

Paper presents an evaluation of sources of plastic pollution aggregated by country. To do so, it models both plastic waste generation and the emission of plastic waste into the environment associated with multiple waste collection processes. These processes are modeled from survey data, where available, or via models taken from the literature, and sometimes by simply using a plausible range of parameters. A material flow analysis is then conducted in which waste is simulated for each metropolitan area and passed through the various models of containment or emission; this is done 5000 times to produce a Monte Carlo estimate of emissions which is then aggregated to the country level. A differentiator for this model is the aggregation of waste models on a per-municipality basis to country or region level.

A key component of this model is a prediction of the distribution of plastic waste generated in each city based on socio-demographic and economic factors. This is produced using a quantile random forests algorithm, trained using data aggregated from four survey sources and then used to simulate waste generated at the municipal level, even where waste data is not available.

The resulting inventory is relevant for upcoming negotiations for a global treaty on plastic pollution.

Commentary:

I think it is, first, important to recognize the complexity of the problem, the modeling effort that went into producing these results, and the timeliness of making such an inventory available. The authors have clearly taken pains to make reasonable modeling choices given highly imperfect, or non-existent, data sources. At some point, of course, the data are so poor that the exercise is fruitless -- I don't think that we are there in this case, but there are some ways the authors could demonstrate that more robustly.

I was specifically asked to review the statistics and the use of random forests approaches employed here. To be frank, amid all the other modeling choices, I doubt that the specifics of the random forests implementation makes a great deal of difference, but see points below.

My first set of comments relate simply to communicating methods and results. Here are things that I am left confused about:

1. Generation of data: for cities for which waste data are available, were data also generated according to the quantile random forest? In some places the paper appears to suggest that these are kept constant:

"Quantile regression random forest models predicted data for remaining global municipalities (those without measured data)"

It isn't clear to me whether the randomness in the MFA is meant to represent variability in production and rates of collection/recycling/litter over time, or whether it is intended to provide an indication of our lack of knowledge. Either way, it seems to me that the data that inform the model are inherently noisy and measured at different times, so that under either interpretation of uncertainty, the MFA should be based entirely on simulated models.

2. "To improve comprehension of proportionality, results aggregated above municipal scale are reported as mean and numbers in brackets are the range unless otherwise specified"

It is not clear to me here what the ranges are over. Municipalities? Simulation runs? I think "simulation runs" is most likely what is intended, which is also what I think is appropriate. But the description could be more precise here.

3. UQ is inconsistently reported, eg municipal plastic waste generated is missing intervals as is unburned debris on lines 56-55 and similar omissions throughout the next few pages. If there are reasons not to report ranges, those should be pointed out.

4. Within the stochastic MFA, it isn't clear at what scale parameters in all of the submodels are simulated. Is a new parameter picked for each municipality? Are the parameters stratified according to LIC/LMC/UMC/HIC? Or according to country? Simulating waste generation by municipality suggests that everything is done on this level, particularly in the absence of other commentary, but it would be good to be precise on this -- possibly some set of pseudo-code could help (admittedly this is challenging to describe over many processes).

Outside of this, we can ask some more methodological questions about the simulation.

1. If my parameters-per-municipality understanding is correct, is simulating all these quantities independently realistic? Let's take, for instance, the primary input variables -- it might be reasonable to expect that municipalities with a high MSW generation rate (Table S20), after accounting for dependent variables, might also have high collection coverage, or that high collection coverage might also correlate with high rates of controlled disposal, even after accounting for independent variables.

Similarly if we might expect municipal rates of informal recycling, littering, etc might be correlated between municipalities within the same country. This should apply under either the variability-over-time or quantifying lack of knowledge interpretation of your simulation.

Given your combination of several data sets, there may not be an easy way to quantify this dependence, but some sensitivity to that should be explored. By making your random numbers more independent, I'd expect your simulation to be doing more averaging and therefore to exhibit smaller ranges than if you induced some form of dependence (see next point).

2. What I had hoped to see in your sensitivity analysis is not an analysis of which waste streams most affected the output of your current model (although that is helpful), but which sources of uncertainty. Eg, what happens if you vary say the location in each of the beta-PERT distributions? If your estimates are strongly dependent on parameters that are only guessed at, or on the specifics of the data sources you chose, this is cause for concern. Personally, I'm not sure that they will be, but that needs to be dealt with.

A little more relevantly, what happens if you induce some dependence, say by using the same random number for all municipalities within a country or geographic region? Or by simulating a single uniform random number for each city and then have them produce all their primary inputs at that quantile (ie to force all the primary inputs to either be high or low together)? I'm sure this can be done more intelligently than I suggest here, but some reasonable attempt to examine the effect of dependence would be helpful.

I think it's also relevant to look at sensitivity to the inclusion/exclusion of data sets used to derive the models, at least in so far as they relate to building the quantile random forest model. Fully removing data sets may hurt performance, but you could down-weight the points (say to 0.5) from each data set in turn to at least see how much the RF is reliant on each data source.

3. How representative are all of your data sources of the final population to which you will apply your model? Some of them clearly skew towards high income countries. This is mentioned in passing, but a table of the values in your data set (say the range, median and standard deviation) and in the municipalities where you apply them would be instructive.

And a couple of miscellaneous points:

1. Your model picks up particular cities or countries as hotspots of plastic waste pollution. It would be helpful to present the evidential basis for this. What are their characteristics that lead to that claim? Does it come from waste production? Or does it come from waste management? If the latter, how are they distinguished so that random forests particularly simulate large values for them?
2. Similarly, where you have notably different estimates from those in the literature, to what do you ascribe these differences and is that particularly due to the focus on more local models?
3. Final quibble in presentation. In Fig. 2 of the main manuscript, the display of panels B, C, and D all emphasize differences between geographic regions. I think that what the authors really want to do, based on their discussion, is to display the proportions in each slice of the pie, but that is not what makes a visual impact and the proportions are difficult to compare across regions, anyway. I think a bar plot of proportions would be far more visually impactful -- if you wanted to also display volume of material you might try varying the width of the bars -- but there are plenty of other alternatives.

Referee #2 (Remarks to the Author):

Review of “A local-to-global emissions inventory of macroplastic pollution”

The research reported in this manuscript and the resulting macroplastic emission inventory represents novel and important contributions to the existing literature and data. The research uses existing municipality-level data of solid waste generation and management as training dataset for a machine learning algorithm (random forest) in order to derive a global, municipality-level dataset on plastic waste generation, management, and emissions. It does this using a very detailed material flow model, which enables the authors to provide unparalleled detail in the emission inventory (see extended data Fig. 1).

While the manuscript, with its 133 pages of SI, is challenging to review, all input data and methods appear to be sound and robust. The material flow model also looks sound. My only concern is with the large amount of data required to populate it. My other concern is the size of the training dataset for the random forest. If I understand correctly, 691 municipal records, some duplicates and many doubtlessly incomplete, were used to generate datasets for 50,702 municipalities. Even discounting duplication, incompleteness, and representativeness of the existing records, that is a coverage of less than 2%. What can be said about the representativeness of the training dataset (input data) and thus the credibility of the results (output data)?

Here are some additional specific questions I have about the manuscript, the methods, and the data:

- Given the considerable uncertainties of all efforts to quantify mismanaged, leaked or emitted plastic waste, the reported results appear to be roughly in line with the existing literature; whereas the authors state that they are not. The authors should try to provide a clearer analysis of similarities and discrepancies with the existing literature.
- It is probably buried somewhere in the 133 page SI, but it is unclear to me 1) which input data have probability density function(pdf), 2) where the pdfs come from, and 3) whether the pdfs are from observations or assumed. This should be explained somewhere where it can be easily found.
- The information contained in extended data Fig.1 needs to be provided earlier, since it is necessary to avoid confusion when reading the results sections.
- The observation that almost all of the plastic waste going to mismanaged disposal is burned is striking. What actual evidence is this based on, and what is the uncertainty of this result? E.g., are the disposal open burning values for Kuwait and Saudi Arabia based actual data of generated by the machine learning algorithm? What about open dumps of apparel waste? Is apparel and other non-packaging plastic waste included in this study?
- Which input datasets report uncollected plastic waste, and how much uncertainty is there for these values? It would seem to me that uncollected waste is not directly observable. Is this data inferred from other input data? This requires clarification.
- I would equally assume that the fraction of uncollected waste that is burned is not directly observable. Again, what is the source of this data, what is its global coverage, and what is its uncertainty? How clear or blurred is the line between uncollected open burned and disposal open burned?

- One more comment regarding the emission type open burning: I would assume that this process involves highly incomplete combustion, to the point of having a significant fraction of plastic waste either remaining or only partially combusted. What is known about this and should this be integrated into the model?

Answering these questions will be key to create confidence in the credibility and robustness of this novel and potentially very important modeling approach and dataset.

Referee #3 (Remarks to the Author):

General Comments

The authors have developed a methodology to create a macroplastic waste inventory that identifies hotspots worldwide from five land-based plastic waste emission sources. They estimate global plastic waste emissions at 52.5 million metric tonnes in 2020, of which 57% is from open burning.

The work is highly significant, of broad general interest and merits publication in a top-tier journal such as Nature. The work provides detailed and evidence-based baseline emission estimates that can inform ongoing negotiations for the international plastic treaty and provides a well-developed set of reference estimates for national and sub-national waste management action plans and source inventories. The authors have also highlighted the importance of sub-national resolution that identifies plastic pollution hotspots (both in absolute value and per capita) and accounts for specific local solid waste management, behavioral, cultural, and socio-economic conditions.

The main text of the article is extremely well-written and concise, and the authors provide a very detailed explanation of their methodology in extensive Supporting Information. The methods appear appropriate, and, for example, the authors have used cross validations extensively in the development of their machine-learning models. I have provided quite a few comments, questions and criticisms below that the authors can consider.

My major criticism of the work is the lack of clear specification of uncertainty ranges on emission estimates at the global and national levels. What are the confidence intervals on these estimates? When comparisons are made between emissions estimates from different sources, or between global regions are they really different considering the confidence ranges? What are the main drivers of uncertainty in this analysis? These questions should be addressed in revisions.

A related important criticism is that the emissions estimates are sometimes implicitly extrapolated to represent a time period much longer than the year 2020 for which they were made. The authors should make the effort to explicitly define the time period over which they think the emissions estimates reported here are valid.

But considered in totality, the article is excellent. It will be interesting to a wide global audience, and it fills an urgent information need. With some (minor, considering the entire scope of this work!) revisions it will become even more clear and useful, and it certainly merits publication in Nature.

Specific Comments

Page 1: “We estimate global plastic waste emissions at 52.5 million metric tonnes...” Be precise here and say “We estimate global plastic water emission in the year 2020 were 52.5 million tonnes...”, ie, clearly specify at this point where they are first stated that these are annual emissions for a 1-year period.

This is actually a big pet peeve of mine that shows up later in the paper... Emission estimates are a flux (here, in Mt / y) for a given year, and I think that clear communication demands that they always be assigned units that clearly convey that they represent a flux.

Page 1: “... and such inventories have already been applied in climate change¹⁸ and air quality¹⁹⁻²¹ fields.”. The IPCC emission inventories are a good analogy. But the air quality inventories perhaps not so much since they serve national or at most regional (ie, UNECE CLRTAP) regulatory instruments. In my opinion the best analogue for this plastic emission inventory is the mercury emission inventory compiled by Pacyna et al. in 2000 in the early days of building political momentum towards the eventual global legally binding agreement on mercury (ie, the Minamata Convention). See: <https://doi.org/10.1016/j.atmosenv.2006.03.041> and <https://doi.org/10.1016/j.atmosenv.2006.03.042> which describe the original Pacyna inventory, and <https://doi.org/10.1002/etc.2374> that discusses updates to the inventory as a basis for Minamata Convention negotiations.)

Figure 1 is a really nice overview of the methodology. If possible it would be nice to draw more attention to the 5 source categories (ie, uncollected waste, littering, collection systems, uncontrolled disposal and rejects). I see they have a small symbol attached to them now, but even more emphasis would be good, especially since they are linked to the red and yellow arrows that show the actual emissions.

Page 3: “... that 52.5 million metric tonnes (Mt) (50.7-54.5 Mt)...” On first reading, the stated range of emissions is very narrow (less than 10% of the total emissions!), and readers will wonder what this range really represents! Is it really meant to be a (95%?) confidence interval, which seems to be suggested for the emissions that the author’s estimate is compared to lower down in this paragraph? I don’t think it can be, since it is so narrow. Later on in the main text most emissions estimates are stated without any accompanying range of uncertainty, and sometimes up to 4 digits of precision are used. As stated in my general comments, the poor level of attention paid to uncertainty in this paper is its major weakness.

Page 4: “... and 22% less than the combined mass of ‘terrestrial’ and ‘aquatic’ pollution (29 Mt 95% CI: 22-39 Mt) reported Lau, et al.⁹ for 2016.” Make it clear here that the Lau et al. estimate has the same dimensions as your estimate (ie, Mt / y) and that it refers to “emissions” and not inventory of plastic pollution. Neither of these things is clear as the sentence is currently written! I suggest... “... and 22% less than the combined emissions of “terrestrial” and “aquatic” plastic pollution in the year 2016 reported by Lau et al.”

Page 4: “Five explanations are suggested for this incongruence with other models:...”. Are the current emissions estimates really “incongruent”? If I understand right the Ryberg et al. and Lau et al. estimates should be compared to the 23 Mt of unburned “debris” estimated by the current model. There is no confidence interval given for the current 23 Mt estimate, but it is very close to the upper range of the confidence interval by Ryberg et al. and within the confidence interval of Lau et al. This all seems quite consistent to me, actually! Perhaps a better framing for this paragraph would be “Our model improves upon earlier estimates and provides new information in five notable ways...”

Page 4: “It is possible that these important and fundamental differences...” I suggest deleting this short paragraph that refers to the “missing plastic” in the oceans. According to my understanding, the “missing plastic” is more than an order of magnitude disagreement between emissions estimates directly to the ocean and inventories. The differences of approximately a factor of 2 between (very!) different inventories described in the previous paragraph are not likely to be a significant part of the explanation.

Page 5: “Overall, we estimate that 56.6 Mt (51.6-62.7 Mt) of municipal solid waste is open burned in India...” Here (and elsewhere throughout the paper!) the authors make some implicit extrapolations through their use of language and the specification of emissions using mass dimensions only (and not mass/time dimensions). Strictly speaking, the authors have estimated that 56.5 Mt of solid waste *was* open burned in India *in 2020*. Of course the authors can justify extrapolating that estimate a bit, and I don’t actually object to them applying it to the current situation, as they seem to want to do. But, I would like to see the authors explicitly state somewhere what time period they think the present analysis can be reasonably extrapolated to (perhaps it is 2015 – 2025?? – the relevant question is: How often does step 6 in Figure 1 need to be carried out?) and then use the correct dimensions (Mt/y) when they quote specific emission estimates.

Figure 2: Another really nice figure. And I like that the caption uses units of Mt/y instead of just Mt! But it would be even better (following my previous comment) if it said “in Mt/y during the period 2015 – 2025”... In Panel C I would prefer to see “Open burning” instead of “Burned” in the caption since this figure is likely to be stolen by others, and out-of-context, “burned” might be misunderstood to mean the plastic itself has been burned (and thus destroyed) rather than that this is plastic released by uncontrolled open burning.

Page 7, top paragraph: In this short paragraph about open burning, I wonder if the authors could specify whether these open burning emissions are directed to air entirely? Or, do they include some emissions to land also? Perhaps this is discussed in the SI somewhere, but this question is very important for modelling the fate of plastic in the environment.

Page 7: “... plastic waste exports from OECD countries to the Global South have plummeted to less than 1.3 Mt·y⁻¹(42),...” This is going to be surprising to many readers, I think. I was surprised! It might help if you added another reference period with quantitative data to this sentence... ie, say “... plastic waste exports from OECD countries to the Global South have plummeted from an average of XXX Mt/y in the period from 1995-2000 to YYY Mt/y in the period from 2015-2020.”

Page 8: Here there is some discussion about large ranges of uncertainty in emission estimates at the municipality level. But I don't see how those large uncertainties propagate to the national and global emissions estimates, and I don't see how such large uncertainties at the municipality level could be consistent with the very narrow range of estimated global emissions back on Page 3. More explanation is needed!

I tried to look into the discussion of uncertainty in Sections S9 and S10 of the Supporting Information, but it is still not clear to me how the high uncertainties in individual processes and estimates at the municipal level have been propagated to the global level. The sensitivity analysis in Section S10 is OK, but sensitivity analysis is most interesting to model developers who want to ensure their code is working right. End-users will be more interested in the product of sensitivity of a model input parameter and its confidence interval, ie, the contribution of each input parameter to variance in the modelled emission estimates.

The most interesting input parameters for further research are those that both have high sensitivity and high uncertainty!

Page 9: In the comparison between Hamburg and Mogadishu it might be more informative to quote the 95% confidence interval of the estimated emissions instead of the maximum and minimum values from the probabilistic modelling. In particular, the lowest value for Mogadishu (0.72 kg/cap/y) looks to be a strong outlier in Figure 3B.

Page 13: "The large mass of waste which is burned..." This paragraph emphasizes that emissions from waste burning have not been given enough attention and that they may even increase as an unintended consequence of global policy action. As also stated above, I would like to see one or two more sentences here that elaborate on whether emissions from waste burning are to air or to land, and if they are to air a statement about the increase spatial scale of the emissions. I also expect waste burning emits plastic in small (microplastic or even nanoplastic) size fractions, which might merit a sentence of discussion.

Small things...

Page 2: "Previous efforts to model global plastic waste emissions and movement through *the* environment..."

Page 7: The acronyms LIC, LMC and UMC are used without explicit definitions.

Page 7: "... emit a similar order of magnitude of plastic waste..." 15.2 and 13.6 are closer together to each other than "a similar order of magnitude". Where are the confidence intervals for these numbers? Are they different from each other within your uncertainties? Maybe not, if uncertainties are highly co-variate, I guess...

Page 12: I am a native English speaker and a scientist engaged in this field, and I had to look up the meaning of "exiguous". Rewrite. Especially since this paper targets a wide general audience.

Author Rebuttals to Initial Comments:

We would like to express our wholehearted thanks for your constructive and detailed comments which have been of great benefit to us in improving the draft manuscript. We are relieved that all three reviewers appear to understand the topic and that they also have sufficient technical understanding to critique our model and method. We appreciate the huge effort involved, especially given the size of the supporting material. We have listed your comments below along with our responses in green text. As you will see, aspects of the model have been redesigned, resulting in a new set of results and substantial amendments to the method in places.

Reviewer 1

Comment 1:

Generation of data: for cities for which waste data are available, were data also generated according to the quantile random forest? In some places the paper appears to suggest that these are kept constant:

"Quantile regression random forest models predicted data for remaining global municipalities (those without measured data)".

It isn't clear to me whether the randomness in the MFA is meant to represent variability in production and rates of collection/recycling/litter over time, or whether it is intended to provide an indication of our lack of knowledge. Either way, it seems to me that the data that inform the model are inherently noisy and measured at different times, so that under either interpretation of uncertainty, the MFA should be based entirely on simulated models.

Seven main solid waste management variables (Primary input data) were predicted for every municipality using the quantile regression random forest. The reviewer is correct in their interpretation, that where measured data were available, we used it to replace predictions. We debated the merits of doing this extensively during the project, not least because of the geopolitical implications.

The randomness of our MFA is designed to reflect the uncertainty of the waste flows according to two general categories:

- 1) **Aleatoric uncertainty:** Observable differences in solid waste management practices between municipalities which may or may not have similar socio-economic characteristics – real world variability.
- 2) **Epistemic uncertainty:** General lack of sufficiently reliable data of data to train the model for a particular set of geographical, socio-economic, or political conditions.

We chose quantile regression random forest as it allows us to quantify both of these types of uncertainty and propagate this into the Monte Carlo probabilistic material flow analysis. We note that as more and standardised data are collected in the future (such as through the Waste Wise Cities Tool, deployed by the UN-Habitat for measuring SDG11.6.1 in cities), we anticipate the epistemic uncertainty to decrease, however aleatoric uncertainty due to varying practices between municipalities will likely still remain.

Following the reviewer's advice, we have re-run the model using purely simulated data. We agree that this is the preferable approach to ensure all data is predicted for the same year (2020). This has affected the model results slightly in a few key areas, notably reducing emissions in some HICs. We have also removed any discussion relating to the data replacement. Please also refer to a new **Section S.9.2.2** in the SI which discusses uncertainty in the model.

Comment 2:

"To improve comprehension of proportionality, results aggregated above municipal scale are reported as mean and numbers in brackets are the range unless otherwise specified"

It is not clear to me here what the ranges are over. Municipalities? Simulation runs? I think "simulation runs" is most likely what is intended, which is also what I think is appropriate. But the description could be more precise here.

Thanks for this – correct they are simulations / iterations. - We've now adjusted the following statement in the method of the main manuscript as follows:

"To improve comprehension of proportionality, results are reported as the mean of all iterations (simulation runs). Numbers in square brackets are the 5th and 95th percentiles of all iterations."

And the first paragraph in the Section on "Global Emissions of Plastic Waste" in the main manuscript:

"(Statistics reported are the arithmetic mean of all iterations (simulation runs). The 5th and 95th percentiles are in square brackets)."

Comment 3:

UQ is inconsistently reported, e.g. municipal plastic waste generated is missing intervals as is unburned debris on lines 56-55 and similar omissions throughout the next few pages. If there are reasons not to report ranges, those should be pointed out.

Following this comment and several others, we now report the 5th and 95th percentiles for results. The percentiles have been directly generated by the model (also see response to Comment 2 above). Very occasionally, we omit some uncertainties in the manuscript where it is overly disruptive to the text and inconsequential to the discussion.

Comment 4:

Within the stochastic MFA, it isn't clear at what scale parameters in all of the submodels are simulated. Is a new parameter picked for each municipality? Are the parameters stratified according to LIC/LMC/UMC/HIC? Or according to country? Simulating waste generation by municipality suggests that everything is done on this level, particularly in the absence of other commentary, but it would be good to be precise on this -- possibly some set of pseudo-code could help (admittedly this is challenging to describe over many processes).

Primary and secondary data inputs are independently sampled from probability density functions (PDF) for each municipality. However, the parameters that define the PDFs are occasionally stratified to higher spatial aggregations (e.g., Income categories) due to absence of widespread municipal level data. This is particularly the case for secondary data input variables where data are often scarce. We have clarified this by adding the spatial scale as a column in **Tables S2** and **S3** in **Section S5** of the SI.

We have also added some narrative to '**Section S1: Methodology summary**' in the SI and as shown below.

*“Primary input variables and secondary input variables within each administrative boundary were assigned a probability distribution from which 5,000 random samples were drawn from each as part of a probabilistic material flow analysis using Monte Carlo simulation. These samples were drawn independently for each municipality; however, the parameters that define the probability distributions were often stratified according to higher spatial aggregations (e.g., income categories) for many of the secondary data inputs due to data constraints (**Section Error! Reference source not found.**)”*

Comment 5:

If my parameters-per-municipality understanding is correct, is simulating all these quantities independently realistic? Let's take, for instance, the primary input variables -- it might be reasonable to expect that municipalities with a high MSW generation rate (Table S20), after accounting for dependent variables, might also have high collection coverage, or that high collection coverage might also correlate with high rates of controlled disposal, even after accounting for independent variables.

Similarly if we might expect municipal rates of informal recycling, littering, etc might be correlated between municipalities within the same country. This should apply under either the variability-over-time or quantifying lack of knowledge interpretation of your simulation.

Given your combination of several data sets, there may not be an easy way to quantify this dependence, but some sensitivity to that should be explored. By making your random numbers more independent, I'd expect your simulation to be doing more averaging and therefore to exhibit smaller ranges than if you induced some form of dependence (see next point).

Thank you for highlighting this, whilst it has been challenging to address, we believe that the reviewer's comment has helped us improve the model.

There are two possible types of dependence relevant to our model: **(1) Intradependence:** Dependence between input variables within a municipality; and **(2) Interdependence:** Dependence of input variables between municipalities.

(1) Intradependence

In our opinion, within municipalities, the primary data inputs used in our model are broadly independent of each other. For example, the reviewer states that municipalities with high waste generation might also have high collection coverage. Whilst this is sometimes the case, we actually found many cases where high waste generation does not infer high waste collection coverage. For example, for the top 20 highest waste generation rates, seven (35%) do not have 100% collection coverage and one reports just 60% collection coverage. Likewise, the reviewer's comment suggests that controlled disposal and collection rate might be linked. We agree that this is the case in some examples, but we also see much evidence of municipalities that have low controlled disposal and a high collection rate and vice versa.

Whilst we acknowledge that there are logical relationships between some variables, and that some dependence may be present between certain input parameters, we argue any dependencies are likely to be weak based on our empirical evidence. Additionally, practical considerations also led us to pursue the assumption of independence. For instance, one approach which would have recognised some of the more complex relationships between input variables would have been to use multivariate random forest. However, during our planning we dismissed this method as we believe it has two major drawbacks: (1) Multivariate random forest doesn't work when datasets are incomplete for a municipality which would have vastly reduced our training data size; and (2) Uncertainty is not generated as per the quantile regression random forest.

As the error in our machine learning results is broadly similar to Velis, et al.¹ who used multivariate random forest, we are reasonably confident that if some form of dependence does exist between variables, it is unlikely to play a substantial role in predictions. We therefore argue it is more realistic to base predictions of primary input data variables solely on the socio-economic conditions of a municipality, rather than any potentially complex relationships to other input variables.

(2) Interdependence (spatial dependency)

Following the reviewer's comment, we have reassessed the possibility of interdependence. Firstly, it is important to note that the responsibility for management of MSW almost always occurs at the municipal level. This is one of the main reasons we believe a bottom-up approach to modelling waste flows is more effective compared to top-down. As such, we think that it is reasonable to expect two neighbouring municipalities to exhibit different levels of service (e.g., collection coverage) depending on independent aspects such as the resources / priorities that they allocate or depending on local socio-economic conditions.

However, we acknowledge that national regulations, policies, or cultural / behavioural practices directly influence solid waste management activities. This means that there is some spatial dependency between municipalities, particularly up to the national level as the reviewer suggest in their comment.

Our model already accounts for dependency (interdependence) between municipalities because many of the independent variables used in the machine learning are either national level variables (e.g. GNI, CPI, SPI, Income category) and constant or similar for every municipality in a country. As a result, the decision trees of the random forest model make similar predictions within countries (i.e. lower the uncertainty / range of predictions of an input variable within a country).

But, following the reviewer's comment, we have identified a potential weakness in the model at the data aggregation stage – the point at which municipal scale data are combined to national, regional or income group level. By assuming independence between municipalities during spatial aggregation, the tails of the distributions effectively cancelled each other out (as explained in **Section S.9.2.2**). This resulted in the uncertainty decreasing as the aggregation levels increased. The low uncertainty has been noted in comments from all reviewers. Importantly, this approach assumed independence between the municipalities, which is not necessarily correct.

We have now re-run the model, changing the way that data are aggregated, using the Iman and Conover² method to induce spatial dependency according to a cross-correlation matrix we specify. This method works by reordering the samples (iterations) to impose the desired level of correlation. For example, positive correlation coefficients would order iterations so that samples with high values tend to fall within the same iteration. It does not impact the values of the samples drawn; therefore, the full uncertainty of each input is still accounted for.

This approach has also been used in other work^{3,4} to aggregate water demand, with us using the same correlation coefficient of 0.5 as these works. We have created a new **Section S.9.2.1** in the SI that explains the spatial aggregation approach. The updated results in the model show that the inclusion of spatial dependency increases the uncertainty range, without impacting on the location (mean).

Comment 6:

What I had hoped to see in your sensitivity analysis is not an analysis of which waste streams most affected the output of your current model (although that is helpful), but which sources of uncertainty. Eg, what happens if you vary say the location in each of the beta-PERT distributions? If your estimates are strongly dependent on parameters that are only guessed at, or on the specifics of the data sources you chose, this is cause for concern. Personally, I'm not sure that they will be, but that needs to be dealt with.

Thank you for this valuable advice. We have expanded our sensitivity analysis to do as the reviewer suggests and test the sensitivity of the model to both input parameters as previously shown (**Section S.10.1** in the SI) and additional sources of uncertainty (i.e. assumptions – **Section S.10.2** in the SI).

To clarify, the Sobol sensitivity analysis we run is a global sensitivity analysis that uses the Monte Carlo approach. This inherently accounts for any uncertainty that we prescribe to inputs based on their PDF. As such, all sensitivity analysis results we show account for the uncertainty of the inputs. However, what we take from the reviewer's comment is that some PDFs have underlying assumptions (e.g., due to low availability or quality of data) and these assumptions should also be tested. To achieve this, we outlined our confidence in the input by introducing now a data pedigree (**Table S38, Section S10.1**). This pedigree is based on the amount and quality of data on which the PDF was constructed. Inputs with a low data pedigree (scores of three or four) are those that we have higher uncertainty over, particularly in relation to the central location (e.g. most-likely value of beta-PERT distributions), or overall range. We ran the sensitivity analysis again using modified PDFs for these uncertain inputs whereby uniform distributions were used throughout to negate any assumptions about central location. Additionally, inputs with the lowest data pedigree scores of four had their range expanded further to test the case that any assumptions we made were incorrect.

We found that the parameters highlighted as influential were almost identical to those of our initial sensitivity analysis, therefore suggesting that outputs were insensitive to any assumptions we made.

Comment 7:

A little more relevantly, what happens if you induce some dependence, say by using the same random number for all municipalities within a country or geographic region? Or by simulating a single uniform random number for each city and then have them produce all their primary inputs at that quantile (ie to force all the primary inputs to either be high or low together)? I'm sure this can be done more intelligently than I suggest here, but some reasonable attempt to examine the effect of dependence would be helpful.

Thank you for these suggestions. Please see Comment 5 for a description of how we introduced spatial dependency between municipalities using the Iman-Conover method. In practice, this works in a similar manner to the quantile method the reviewer describes in their comment.

Comment 8:

I think it's also relevant to look at sensitivity to the inclusion/exclusion of data sets used to derive the models, at least in so far as they relate to building the quantile random forest model. Fully removing data sets may hurt performance, but you could down-weight the points (say to 0.5) from each data set in turn to at least see how much the RF is reliant on each data source.

Thank you for this comment. We did not consider the sensitivity of the random forest models to the inclusion/exclusion of datasets, because we believe the cross-validation stage (10-fold with 5 repeats) automatically ensures the predictions are not overly sensitive to the selection of certain data (**Section S.7**). For example, during cross-validation, the training data are randomly split into 10 partitions (folds), with the model trained on nine of the folds and tested on the remaining one (validation fold). This is then repeated for each fold, iterating the fold used as the validation fold. This whole process is then repeated five times to ensure the results are not sensitive to the initial random sampling of folds, with the final performance calculated as the mean from each iteration and repeat. As such, this extensive cross-validation process is designed to train and test the performance of the random forest using different subsets of data. Although we agree that understanding the sensitivity of the random forest models to the exclusion of datasets would be potentially interesting, we believe that omitting this aspect does not affect the validity of the model.

Comment 9:

How representative are all of your data sources of the final population to which you will apply your model? Some of them clearly skew towards high income countries. This is mentioned in passing, but a table of the values in your data set (say the range, median and standard deviation) and in the municipalities where you apply them would be instructive.

The primary Input data used by the model cover 12.2% of the global population, which on an income group basis is: LIC: 12.0%, LMC: 11.4%, UMC: 13.5%, HIC: 11.2%. We cover this in **Section S.6.7** of the Supplementary Information and provide several charts in **Fig. S10**. We have also added a sentence to the method in the main manuscript explaining how representative the input data are and where to find further information in the SI as shown below:

*“Our primary input data represent 12.2% of the 2015 global population, spanning each of the World Bank income categories (LIC: 12.0%, LMC: 11.4%, UMC: 13.5%, HIC: 11.2%). Further discussion on the representativeness of our input data is presented in **Section S.6.7**.”*

Comment 10:

Your model picks up particular cities or countries as hotspots of plastic waste pollution. It would be helpful to present the evidential basis for this. What are their characteristics that lead to that claim? Does it come from waste production? Or does it come from waste management? If the latter, how are they distinguished so that random forests particularly simulate large values for them?

Socioeconomics are used by the model to simulate waste management characteristics, performance, and quality in municipalities (**Section S.7**) based on our primary input data (**Section S.6**). These simulated data 'feed' the MFA (**Section S.9**). Where data are not available, because they are unmeasured or uncertain (refer to uncertainty section), we created sub-models (**Section S.8**). The combination of the 81 processes modelled results in plastic waste emissions that can be ranked to identify hotspots.

Hotspots of plastic emissions are discussed in the manuscript on both an absolute and per capita basis. Per capita hotspots are discussed in the section entitled '**Per capita emission hotspots**' and are a result of both high plastic waste generation rates and poor solid waste management. For example, our sensitivity analysis (**Section S.10**) highlights that plastic waste generation (proportion in MSW) and collection coverage are the main parameters that influence the magnitude of emissions. We further discuss the main sources that lead to these hotspots within the section '**Sources of plastic emissions**'. Here, we explain that in the Global South, lack of waste collection is the primary driver, whereas in the Global North, litter and the collection system proportionally contribute the most material. As we explain, the 1.5 billion people without waste collection must 'self-manage' their waste.

The hotspots of absolute emissions are also dependent on high plastic generation and poor waste management, however, as would be expected; they are also strongly driven by population. For example, as explained in the main manuscript section entitled '**Plastic emission hotspots outlook**', despite China's advances in waste management provision over recent decades, it is still in the top 5 players on an absolute basis because there are 1.3 billion people, therefore even small gaps in collection coverage can lead to high absolute emissions. We explain in the section that our work differs from other models which use Chinese data from the early 2000s.

Comment 11:

Similarly, where you have notably different estimates from those in the literature, to what do you ascribe these differences and is that particularly due to the focus on more local models?

As discussed in Comment 10 above, in the section of the main manuscript entitled '**Plastic emission hotspots outlook**', we discuss how our model differs from previous estimates of emissions from China because we have acquired much more reliable data. Previous models used data for China from What a Waste 1.0⁵, the source for which was a decade older or even the Waste Atlas⁶ – an impressive but completely un-curated dataset which does not undergo quality assurance. In the same section, we have also improved the comparison of the overall model with two comparable global models in the second paragraph – hopefully improving comprehensibility. The fact that our model is a bottom-up municipal model and is created based around conceptualisation of emissions are also a key distinction from previous models; however, it is unclear whether it was this higher spatial resolution or emission mechanisms conceptualisation or improved and updated datasets that led to the largest differences – likely all contribute.

We have also found two other places where we can expand on our comparisons. (1) In the "**Sources of plastic emissions**" section of the main manuscript, we have now expanded on our narrative towards the end of the first paragraph, comparing our estimated emissions from uncontrolled disposal with Lau et al. 2020; and (2) At the end of the main manuscript section entitled "**Global emissions of plastic waste**", we compare the open burning estimate from Lau et al. with our own and explain why we think it is different.

Comment 12:

Final quibble in presentation. In Fig. 2 of the main manuscript, the display of panels B, C, and D all emphasize differences between geographic regions. I think that what the authors really want to do, based on their discussion, is to display the proportions in each slice of the pie, but that is not what makes a visual impact and the proportions are difficult to compare across regions, anyway. I think a bar plot of proportions would be far more visually impactful -- if you wanted to also display volume of material you might try varying the width of the bars -- but there are plenty of other alternatives.

Thanks – we debated these charts extensively during the project – in fact, earlier versions were bar charts as the reviewer suggests! Based on the reviewer's comment, we have now reverted to stacked bar charts, which we hope the reviewer agrees, present the data more clearly.

Reviewer 2

Comment 1:

My only concern is with the large amount of data required to populate it.

We agree that the data used to populate the model were extensive – we believe that our efforts to build the largest quality assured municipal scale database have addressed this, combined with our creation of sub-models, supplemented with plausibility checks. We anticipate that the UN efforts to improve this data paucity with the Waste Wise Cities Tool⁷ will make future iterations even better.

Comment 2:

My other concern is the size of the training dataset for the random forest. If I understand correctly, 691 municipal records, some duplicates and many doubtlessly incomplete, were used to generate datasets for 50,702 municipalities. Even discounting duplication, incompleteness, and representativeness of the existing records, that is a coverage of less than 2%. What can be said about the representativeness of the training dataset (input data) and thus the credibility of the results (output data)?

To clarify, we began the modelling process with 704 municipal data records, which after cleaning, harmonising, and duplicate removal reduced down to 553 records with 2,688 data points (see **Fig.S11** in the SI). We agree that the number of quality assured municipal data points available is small compared to the number of global municipalities; this is a substantial challenge for our model and all previous top-down models, the data for which are either derived from the same local data paucity, or highly uncertain estimates. In **Section S.6.7** in the supplementary Information, we discuss this data paucity, but we also explain that although our dataset only covers 2% of municipalities, that those municipalities represent >12% of the global population. Following the reviewer's comment and one from another reviewer, we have now added two sentences at the end of the 2nd paragraph of the method summary in the main manuscript which explain this, and which direct readers to **Section S.6.7** in the supplementary Information.

Despite the low coverage of absolute number of municipalities, we do not believe this affects the credibility of the results for three reasons: (1) We paid great attention to ensure the data we used were reliable at the expense of having a larger dataset. We believe this trade-off is justified given the generally poor quality of solid waste management data available. We argue that having fewer data points with high reliability is preferable to having more data points with lower reliability. We also note that the dataset we use is the most comprehensive to date. (2) Many of the municipalities we predict for have very similar socio-economic conditions (as represented by the independent variables). The random forest would treat these municipalities in a similar manner; therefore, we argue that practically, there are fewer 'unique' municipalities that the machine learning predicts for. (3) Given the way the random forest works, we believe the most crucial aspect is that our training data covers a wide variety of socio-economic situations so that these situations can be predicted. Given the large geographical spread of data (**Figure S9** in the SI), and socio-economic realities that these cover, we believe this has been achieved. However, we acknowledge that rural areas in particular are a socio-economic condition that is under-represented by the available input data (which we have taken substantial steps to correct and account for – **Section S.9.1.2**), and advocate for more data collection in these areas (**Section S.10**).

Comment 3:

Given the considerable uncertainties of all efforts to quantify mismanaged, leaked or emitted plastic waste, the reported results appear to be roughly in line with the existing literature; whereas the authors state that they are not. The authors should try to provide a clearer analysis of similarities and discrepancies with the existing literature.

Thanks very much for this comment which is similar to the observations made by Reviewer 3 (Comments 7 and 8). We have rewritten this section in an attempt to make it clearer and hope the reviewer agrees that our new expanded narrative explains more clearly how our model compares to other work.

Comment 4:

It is probably buried somewhere in the 133 page SI, but it is unclear to me 1) which input data have probability density function (pdfs), 2) where the pdfs come from, and 3) whether the pdfs are from observations or assumed. This should be explained somewhere where it can be easily found.

The details which the reviewer requires on PDFs are spread throughout **Section S.8** in the SI for *secondary data* inputs and from the machine learning (**Section S.7**) for the *primary data inputs*. These PDFs are also summarised in their entirety in **Supplementary Data Table 6**, which will be uploaded to a repository on submission. We were unable to upload this with the other data files due to file size limitations. If the reviewer requires this in advance, it can be requested from the editor. Following the reviewer's comment, and to make this clearer to the reader we have taken two further actions: **(1)** We have explicitly explained in the introduction to **Section S.8** where the information can be found in the following subsections and where the summarised table is; and **(2)** We have done the same in the method at the back of the main manuscript.

Comment 5:

The information contained in extended data Fig.1 needs to be provided earlier, since it is necessary to avoid confusion when reading the results sections.

We had already referenced **Extended data Fig.1** in the first paragraph of the introduction. However, our terminology and system definitions appear to have confused all three reviewers – so thanks for pointing this out. We have now substantially re-written the section to explain how we define emissions, and clarified what is meant by open burning.

Comment 6:

The observation that almost all of the plastic waste going to mismanaged disposal is burned is striking. What actual evidence is this based on, and what is the uncertainty of this result? E.g., are the disposal open burning values for Kuwait and Saudi Arabia based actual data of generated by the machine learning algorithm?

Thank you for this comment. We have two responses:

- (1) We believe that this comment relates to two possible statements in the Section of the main manuscript entitled Sources of plastic emissions:
 - *“Here, we show that plastic waste emissions from uncontrolled land disposal sites (dumpsites), whilst important, contribute (.....) of global plastic waste emissions, of which 98% (wt.) is open burned. This means that just (.....) is emitted from land disposal sites as debris, substantially less than has been inferred elsewhere⁸.”*
 - *“As with the Global South, most emissions from disposal sites in HICs (98% wt.) are from open burning, meaning that plastic debris emissions are comparatively negligible”.*

These two statements do not say that most plastic sent to unmanaged disposal is burned, rather that the majority of emissions from uncontrolled disposal are from open burning – what this means is that proportionally, very little is debris.

- (2) The data suggesting uncontrolled disposal in Kuwait and Saudi came from the What a Waste 2.0 cities dataset. Originally, we replaced our machine learning predictions with actual data for any municipalities where it existed (originally detailed in **Section S.7.4.1**). However, following Comment 1 from Reviewer 1 advising us to avoid this data replacement, we have now adjusted the model to include only simulated data making the statements about Kuwait and Saudi moot given their predictions are for controlled disposal (for which we assume no open burning). We have now removed the paragraph around Kuwait and Saudi.

Comment 7:

What about open dumps of apparel waste? Is apparel and other non-packaging plastic waste included in this study?

Textiles are excluded as described in the first paragraph of the Scope presented in **Supplementary Section S.2**. However, as the reviewer is also interested, others may want to see this clearly explained in the main manuscript. Therefore, we have added the following sentence to the method at the end of the main manuscript:

“We also exclude textiles, sea-based sources of plastic pollution and waste electrical and electronic equipment”

Comment 8:

Which input datasets report uncollected plastic waste, and how much uncertainty is there for these values? It would seem to me that uncollected waste is not directly observable. Is this data inferred from other input data? This requires clarification.

A valid point, given the influence that uncollected waste has on the model. Firstly, the reviewer is correct that uncollected waste is challenging to measure – as with many parts of unmanaged system, there is rarely compulsion or motivation to measure these aspects which occur outside of the formal system. In the case of uncollected waste, we model this by taking the complement of the collection coverage (i.e., households without collection services) (**Section S.4.1**). These data are mainly derived from estimates of number of households or population served as a proportion of total households or population. Both are usually known to some extent and are reported in all four of the international datasets (see throughout **Section S.6**). Typically, they do not provide uncertainty – the type of reporter has little compulsion to do so. Instead, we train the Random Forest model, and it provides the uncertainty for us, as described in **Section S.5** and **Table S2**. Given we have lots of data on collection coverage, we are reasonably confident in these predictions (see data pedigree of **Section S10.1**).

Comment 9:

I would equally assume that the fraction of uncollected waste that is burned is not directly observable. Again, what is the source of this data, what is its global coverage, and what is its uncertainty? How clear or blurred is the line between uncollected open burned and disposal open burned?

The reviewer is correct that open burning is highly challenging to quantify. Actually a few papers have been published recently which offer methods for quantification, but they are highly localised and carry a lot of uncertainty. Instead, we used activity data obtained from census and surveys for 44 countries that asks about people's disposal habits. We explain and discuss how we obtained and assessed these data in **Section S.8.11**.

Open burning on disposal sites is much more challenging to quantify as there is limited data or methods to assess this. We explain how we obtained data in **Section S.8.11.3** and highlight this as a weakness in the model in the sensitivity analysis section.

Our model treats open burning of uncollected waste and open burning at uncontrolled disposal sites as two distinct processes. Our process for open burning at uncontrolled disposal sites relates to formal dumpsites after collection by authorities (see MFA diagram of **Figure S5**). We acknowledge that some uncollected waste may be dumped in informal dumpsites and open burned, however, this is incorporated within the uncollected waste open burning process in our model.

Comment 10:

One more comment regarding the emission type open burning: I would assume that this process involves highly incomplete combustion, to the point of having a significant fraction of plastic waste either remaining or only partially combusted. What is known about this and should this be integrated into the model?

This is a valid point and a fantastic topic for future research. Undoubtedly microplastics will be generated following the incomplete combustion of macroplastics in open uncontrolled fires. Some evidence exists for this, of studies of incinerator bottom ash. As we are not accounting for microplastics here, these were not considered in our mass balance.

Reviewer 3

Comment 1:

My major criticism of the work is the lack of clear specification of uncertainty ranges on emission estimates at the global and national levels. What are the confidence intervals on these estimates? When comparisons are made between emissions estimates from different sources, or between global regions are they really different considering the confidence ranges? What are the main drivers of uncertainty in this analysis? These questions should be addressed in revisions.

Thank you. We agree that uncertainty was not explained well. We have done the following to address your comments and several others:

(1) We now reported uncertainty against every value for which it has been calculated throughout the main manuscript. To clarify, we do not use confidence intervals, but instead report the mean and percentiles of simulation runs (iterations). Previously we reported the range, but we now report the 5th and 95th percentile in square brackets. This is explained the first time it is used and also in the method at the end of the main manuscript;

(2) We have changed the way that uncertainty is aggregated from municipal to national and regional level. This new approach is explained in a **Section S.9.2.1** of the SI;

(3) We have created a new section in the SI – **Section S.9.2.2** which summarises our approach to uncertainty; and

(4) We have expanded the sensitivity analysis section which aims to assess the main drivers of uncertainty in the analysis. In this section we now include a table which scores model parameters according to a data pedigree.

Regarding the reviewer's comment about whether we can be confident in our comparisons when accounting for uncertainty, the data in **Supplementary Table 4** show that most UN regions uncertainty ranges do not overlap. The same is the case for many of the other major findings of our work (e.g., uncollected waste being the largest source, India having the largest emission etc.). As such, we believe the conclusions we present are valid given our uncertainty.

Comment 2:

A related important criticism is that the emissions estimates are sometimes implicitly extrapolated to represent a time period much longer than the year 2020 for which they were made. The authors should make the effort to explicitly define the time period over which they think the emissions estimates reported here are valid.

We are not aware that our narrative suggest that results are implicitly represented over a much longer time period. But following this comment, Comment 3 and 10, we have now restated the 2020 reference year in several places as well as in the Method and SI.

Comment 3:

Page 1: “We estimate global plastic waste emissions at 52.5 million metric tonnes...” Be precise here and say “We estimate global plastic water emission in the year 2020 were 52.5 million tonnes...”, ie, clearly specify at this point where they are first stated that these are annual emissions for a 1-year period.

This is actually a big pet peeve of mine that shows up later in the paper... Emission estimates are a flux (here, in Mt / y) for a given year, and I think that clear communication demands that they always be assigned units that clearly convey that they represent a flux.

Yes, fully agree – please see response to Comment 2 and 10 – we have now explicitly stated the year and assigned correct units.

Comment 4:

Page 1: “... and such inventories have already been applied in climate change¹⁸ and air quality¹⁹⁻²¹ fields.”. The IPCC emission inventories are a good analogy. But the air quality inventories perhaps not so much since they serve national or at most regional (ie, UNECE CLRTAP) regulatory instruments. In my opinion the best analogue for this plastic emission inventory is the mercury emission inventory compiled by Pacyna et al. in 2000 in the early days of building political momentum towards the eventual global legally binding agreement on mercury (ie, the Minamata Convention).

See: <https://doi.org/10.1016/j.atmosenv.2006.03.041> and <https://doi.org/10.1016/j.atmosenv.2006.03.042> which describe the original Pacyna inventory, and <https://doi.org/10.1002/etc.2374> that discusses updates to the inventory as a basis for Minamata Convention negotiations.)

We are very grateful for this and have added these references and the associated narrative (pasted below):

A global plastic pollution emission inventory has been suggested as critical to the success of the Plastics Treaty¹⁷ and such inventories have already been applied in the climate change field¹⁸ and as early evidence for a global legally binding agreement on mercury^{19,20}; eventually the Minamata Convention²¹.

Comment 5:

Figure 1 is a really nice overview of the methodology. If possible it would be nice to draw more attention to the 5 source categories (ie, uncollected waste, littering, collection systems, uncontrolled disposal and rejects). I see they have a small symbol attached to them now, but even more emphasis would be good, especially since they are linked to the red and yellow arrows that show the actual emissions.

We're really pleased that the diagram made sense and that the reviewer made the connection with the arrows. We have worked with the illustrator on a few options following the reviewer's comment. We have tried colouring the boxes, the badges and both. In each case, the additional colour made the overall diagram look too busy – detracting from the various messages we try to convey. So instead, we have opted to simply enlarge the badges – we hope the reviewer agrees that this change highlights the sources sufficiently.

Comment 6:

Page 3: "... that 52.5 million metric tonnes (Mt) (50.7-54.5 Mt)..." On first reading, the stated range of emissions is very narrow (less than 10% of the total emissions!), and readers will wonder what this range really represents! Is it really meant to be a (95%?) confidence interval, which seems to be suggested for the emissions that the author's estimate is compared to lower down in this paragraph? I don't think it can be, since it is so narrow. Later on in the main text most emissions estimates are stated without any accompanying range of uncertainty, and sometimes up to 4 digits of precision are used. As stated in my general comments, the poor level of attention paid to uncertainty in this paper is its major weakness.

Thank you for this insightful comment – we understand why the reviewer was concerned with the small uncertainty ranges reported previously. We explain in the method at the end of the main manuscript that these numbers represented the range across simulation runs (iterations). In retrospect, we should have put this upfront, as we have now done. However, following the reviewer's comment and comments from both other reviewers, we have changed the way we calculate and report uncertainty - Please refer to our response to Comment 1. For example, we assume spatial dependency during the aggregation stage, which in turn has widened our uncertainty range. Full details of this aggregation and reasoning behind why the uncertainty reported decreases at aggregated levels is discussed in the new section on uncertainty in the SI (**Section S.9.2.2**).

Comment 7:

Page 4: "... and 22% less than the combined mass of 'terrestrial' and 'aquatic' pollution (29 Mt 95% CI: 22-39 Mt) reported Lau, et al.9 for 2016." Make it clear here that the Lau et al. estimate has the same dimensions as your estimate (ie, Mt / y) and that it refers to "emissions" and not inventory of plastic pollution. Neither of these things is clear as the sentence is currently written! I suggest... "... and 22% less than the combined emissions of "terrestrial" and "aquatic" plastic pollution in the year 2016 reported by Lau et al."

Thank you for highlighting. Comparisons are challenging with all of the current models, we know this well, and we have good reasons not describe them as incongruent, but it's clear we haven't been able to communicate this well. Based on this Comment (7), Comment 8 below, and Reviewer 2 Comment 3, we have done the following:

- We have followed the reviewer's advice and rephrased the five improvements – we have restructured and put this part of the narrative first which seems a more logical order
- We have included a sentence which briefly explains the ways that all the other models report 'emissions'
- We have explained the narrative around the comparisons with Lau et al. and Ryberg et al., explaining in detail the differences between their estimates and ours – clarifying what we mean by the 'incongruence' (we acknowledge this was unclear before)

Comment 8:

Page 4: "Five explanations are suggested for this incongruence with other models:..." Are the current emissions estimates really "incongruent"??

If I understand right the Ryberg et al. and Lau et al. estimates should be compared to the 23 Mt of unburned "debris" estimated by the current model. There is no confidence interval given for the current 23 Mt estimate, but it is very close to the upper range of the confidence interval by Ryberg et al. and within the confidence interval of Lau et al.

This all seems quite consistent to me, actually! Perhaps a better framing for this paragraph would be "Our model improves upon earlier estimates and provides new information in five notable ways..."

Please see Comment 7 above.

Comment 9:

Page 4: "It is possible that these important and fundamental differences..." I suggest deleting this short paragraph that refers to the "missing plastic" in the oceans. According to my understanding, the "missing plastic" is more than an order of magnitude disagreement between emissions estimates directly to the ocean and inventories. The differences of approximately a factor of 2 between (very!) different inventories described in the previous paragraph are not likely to be a significant part of the explanation.

Thank you – we agree and have deleted it.

Comment 10:

Page 5: “Overall, we estimate that 56.6 Mt (51.6-62.7 Mt) of municipal solid waste is open burned in India...” Here (and elsewhere throughout the paper!) the authors make some implicit extrapolations through their use of language and the specification of emissions using mass dimensions only (and not mass/time dimensions). Strictly speaking, the authors have estimated that 56.5 Mt of solid waste *was* open burned in India *in 2020*. Of course the authors can justify extrapolating that estimate a bit, and I don’t actually object to them applying it to the current situation, as they seem to want to do. But, I would like to see the authors explicitly state somewhere what time period they think the present analysis can be reasonably extrapolated to (perhaps it is 2015 – 2025?? – the relevant question is: How often does step 6 in Figure 1 need to be carried out?) and then use the correct dimensions (Mt/y) when they quote specific emission estimates.

The reviewer is correct that our model uses 2020 as the reference year. We debated about the use of Mt versus $\text{Mt}\cdot\text{y}^{-1}$ and had originally decided to drop the $\cdot\text{y}^{-1}$ because we have already explicitly stated in the first results paragraph, the method and SI that we refer to 2020. However, in hindsight we agree with the reviewer that this may not be immediately clear so for clarity we have now done the following: (1) we have added the time dimension to every number throughout as per the reviewer’s suggestion; and (2) we have made clear in the captions to the figures that we refer to the year 2020.

As 2020 was our reference year, we have built the narrative and model around it so would prefer not to comment on whether the data are extrapolatable to other years.

Step 6 is not something which needs to be carried out with any particular frequency, it was meant to show that the model is designed to be updated with data collected by the UN as part of the Waste Wise Cities initiative. Following the reviewer’s comment, we have decided that this is a distraction from the message we want to convey so we have decided to remove it completely.

Comment 11:

Figure 2: Another really nice figure. And I like that the caption uses units of Mt/y instead of just Mt! But it would be even better (following my previous comment) if it said “in Mt/y during the period 2015 – 2025” ... In Panel C I would prefer to see “Open burning” instead of “Burned” in the caption since this figure is likely to be stolen by others, and out-of-context, “burned” might be misunderstood to mean the plastic itself has been burned (and thus destroyed) rather than that this is plastic released by uncontrolled open burning.

Thank you:

- 1) We have now clarified in all captions that we refer to 2020
- 2) All numbers, captions, and figures include time dimensions
- 3) Changed ‘burned’ to ‘open burning’
- 4) For clarification, when we use the term open burning, we do refer to material which has been subjected to open burning – not the emissions of gas, liquid or solid from the open burning process. Somehow, we have not communicated that well. We have now strengthened our definitions of ‘emissions’, ‘debris’, and ‘open burning’ in the second paragraph of the introduction and hope that this is all clear now.

Comment 12:

Page 7, top paragraph: In this short paragraph about open burning, I wonder if the authors could specify whether these open burning emissions are directed to air entirely? Or, do they include some emissions to land also? Perhaps this is discussed in the SI somewhere, but this question is very important for modelling the fate of plastic in the environment.

Please see point 4 in Comment 11 above.

Comment 13:

Page 7: "... plastic waste exports from OECD countries to the Global South have plummeted to less than 1.3 Mt·y⁻¹(42),..." This is going to be surprising to many readers, I think. I was surprised! It might help if you added another reference period with quantitative data to this sentence... ie, say "... plastic waste exports from OECD countries to the Global South have plummeted from an average of XXX Mt/y in the period from 1995-2000 to YYY Mt/y in the period from 2015-2020."

Following the reviewer's suggestion to compare contemporary trade with historical highs, we have taken some time to review our assertions about transboundary trade – if the reviewer is surprised, then others will be too. We don't think it makes sense to take an average in this case, because there are clear trends driven by specific policy changes - for example the Chinese import ban of 2018. Therefore, we have decided to go for a comparison with 2017, the last year of historical highs before the Chinese import ban, after which there was a rapid drop and then a steady downward trend in trade.

As we discussed the reviewer's comment, we realised that our assertion about trade to the 'Global South' was slightly misleading. Actually, the analysis we cited was for trade from OECD to non-OECD countries. Importantly, Turkey, a significant destination for plastic waste in recent years, and one where there is evidence of its mismanagement, is an OECD country, so its omission from our analysis is misleading.

Therefore, we now provide a new analysis, comparing the most recent year of complete Comtrade data (2022) with 2017. We present the data in **Section S.2** and have slightly adjusted the narrative. The overall story is the same in that exports are not important to the overall message, even with Turkey, but we believe our assessment is now more robust and substantiated.

Comment 14:

Page 8: Here there is some discussion about large ranges of uncertainty in emission estimates at the municipality level. But I don't see how those large uncertainties propagate to the national and global emissions estimates, and I don't see how such large uncertainties at the municipality level could be consistent with the very narrow range of estimated global emissions back on Page 3. More explanation is needed! I tried to look into the discussion of uncertainty in Sections S9 and S10 of the Supporting Information, but it is still not clear to me how the high uncertainties in individual processes and estimates at the municipal level have been propagated to the global level.

Yes – agreed – please see our responses to Comment 1 and 6.

Comment 15:

The sensitivity analysis in Section S10 is OK, but sensitivity analysis is most interesting to model developers who want to ensure their code is working right. End-users will be more interested in the product of sensitivity of a model input parameter and its confidence interval, ie, the contribution of each input parameter to variance in the modelled emission estimates. The most interesting input parameters for further research are those that both have high sensitivity and high uncertainty!

Thank you. We agree that end-users are most interested in the how each input contributes to variance in the model. This type of sensitivity analysis is what we show in **Section S.10.1** using the variance-based Sobol model, which is a global sensitivity analysis that includes both first order (direct effects), and total effects (including all higher-order interactions). We conclude that inputs relating to (plastic) waste generation and collection are those most influential to the model outputs, however, given these are also the inputs on which we have the most reliable data, we can be confident in the results. To demonstrate this more, we have included a data pedigree table (**Table S38**) to show our confidence around each input. Based on the reviewer's comment, we also now draw more attention to the parameters that have high sensitivity and high uncertainty:

Section S.10.1 - "On the other hand, we acknowledge that some of the lower ranked influential input parameters were allocated poor data pedigrees, for example, open burning at uncontrolled disposal sites (C8) and the settlement typology correction multipliers for waste generation (RC_tP1_{pc}) and plastic (RC_C0). Likewise, in HICs, no/limited data was available for the efficiency of street sweepings (S), emissions from the collection system (C3i) and the total littering rate (LT). As such, we recommend data collection efforts in the future to focus on improving the availability and quality of data surrounding these input parameters, particularly in the HIC context where they are more influential."

In addition to the above, we have also expanded the sensitivity analysis further to show the sensitivity of model outputs to any assumptions we made about the location or range of uncertain input parameters (**Section S.10.2**). The results are largely similar to the first sensitivity analysis, showing that they do not substantially influence the model results.

Comment 16:

Page 9: In the comparison between Hamburg and Mogadishu it might be more informative to quote the 95% confidence interval of the estimated emissions instead of the maximum and minimum values from the probabilistic modelling. In particular, the lowest value for Mogadishu (0.72 kg/cap/y) looks to be a strong outlier in Figure 3B.

As explained in earlier comments, we have overhauled the way uncertainty is calculated and reported across the whole model. We now report 5th and 95th percentiles for the cities in **Figure 3B**.

Comment 17:

Page 13: “The large mass of waste which is burned...” This paragraph emphasizes that emissions from waste burning have not been given enough attention and that they may even increase as an unintended consequence of global policy action. As also stated above, I would like to see one or two more sentences here that elaborate on whether emissions from waste burning are to air or to land, and if they are to air a statement about the increase spatial scale of the emissions. I also expect waste burning emits plastic in small (microplastic or even nanoplastic) size fractions, which might merit a sentence of discussion.

Apologies - we have confused the reviewer by not explaining clearly enough what is meant by open burning – please refer to the second paragraph of the introduction which has now been amended to explain our terminology more clearly and to our response to Comments 11 and 12. We prefer not to refer to micro or nano plastics which are out of scope here and which we believe may add confusion to an already complex narrative.

Comment 18:

Page 2: “Previous efforts to model global plastic waste emissions and movement through *the* environment...”

Corrected – thank you.

Comment 19:

Page 7: The acronyms LIC, LMC and UMC are used without explicit definitions.

Corrected – thank you.

Comment 20:

Page 7: “... emit a similar order of magnitude of plastic waste...” 15.2 and 13.6 are closer together to each other than “a similar order of magnitude”. Where are the confidence intervals for these numbers? Are they different from each other within your uncertainties? Maybe not, if uncertainties are highly co-variate, I guess...

We have changed ‘similar order of magnitude’ to ‘amount’ and added percentiles to all numbers in square brackets.

Comment 21:

Page 12: I am a native English speaker and a scientist engaged in this field, and I had to look up the meaning of “exiguous”. Rewrite. Especially since this paper targets a wide general audience.

Relevant author hangs head in shame for pompous language – replaced with ‘very small’ – sorry.

References

- 1 Velis, C. A., Wilson, D. C., Gavish, Y., Grimes, S. M. & Whiteman, A. Socio-economic development drives solid waste management performance in cities: A global analysis using machine learning. *Sci. Total Environ.* **872**, 161913 (2023).
- 2 Iman, R. L. & Conover, W. J. A distribution-free approach to inducing rank correlation among input variables. *Communications in Statistics - Simulation and Computation* **11**, 311-334 (1982).
- 3 Tolson Bryan, A., Maier Holger, R., Simpson Angus, R. & Lence Barbara, J. Genetic Algorithms for Reliability-Based Optimization of Water Distribution Systems. *Journal of Water Resources Planning and Management* **130**, 63-72 (2004).
- 4 Magini, R., Boniforti, M. A. & Guercio, R. Generating Scenarios of Cross-Correlated Demands for Modelling Water Distribution Networks. *Water* **11** (2019).
- 5 Hoornweg, D. & Bhada-Tata, P. What a waste: A global review of solid waste management. https://siteresources.worldbank.org/INTURBANDEVELOPMENT/Resources/336387-1334852610766/What_a_Waste2012_Final.pdf (Urban Development & Local Government Unit - World Bank, Washington, DC, USA, 2012).
- 6 Waste Atlas. Waste Atlas <http://www.atlas.d-waste.com/> (2022).
- 7 UN-Habitat. Waste Wise Cities Tool: Step by Step Guide to Assess a City's Municipal Solid Waste Management Performance through SDG indicator 11.6.1 Monitoring. <https://unhabitat.org/sites/default/files/2021/02/Waste%20wise%20cities%20tool%20-%20EN%203.pdf> (UN-Habitat, Nairobi, Kenya, 2021).
- 8 Lau, W. W. Y. *et al.* Evaluating scenarios toward zero plastic pollution. *Science* **369**, 1455-1461 (2020).

Reviewer Reports on the First Revision:

Referee #1 (Remarks to the Author):

I would like to commend the authors on the work they put into this revision. My most substantial concern -- dependence between model outputs -- has been addressed in a reasonable manner.

I dislike the terms epistemic and aleatoric uncertainty, partly because they're frankly pretentious but mostly because their mapping onto "uncertainty about model predictions" and "additional uncertainty about the values of a new data point" doesn't really match their original definitions. This does appear to be their current use, however, and instead I'll confine myself to noting that, in contrast to the discussion on p. 101 of the supplement, correlation between municipalities can also be induced due to epistemic uncertainty about modeling waste collection processes.

My purpose in this bit of nit-picking is to query the authors choices not to include uncertainty about controlled dispersal of MSW (Supplement, In 2022, pg 93). This may be bimodal, but quantile random forests including both 0% and 100% for a given municipality is surely a reflection of epistemic uncertainty. If you claim to be accounting for both sources of uncertainty is it reasonable to leave this out? You might claim that this is a failure of the model in that it does not correspond to what we know about municipalities where we have reliable data, but then why should I trust it for municipalities that I'm simulating?

A second comment is that it is helpful to be precise about the purposes of your sensitivity analysis and how that maps onto how you conducted it. The original analysis used Sobol indices to examine the impact of inputs on outputs, with the goal of understanding to what extent poorly-measured quantities were associated with variability in outputs.

The additional sensitivity analysis asks how the sensitivity analysis changes if we replace Beta-PERT distributions with uniforms. This is helpful in that it does not change what we think is important to measure better. That said, my original question had been about how your modeling choices impact the point estimates and uncertainty of your practical estimates -- does the act of changing one Beta-PERT to uniform change your estimates for total plastic waste, for example? To do this you could look at Sobol indices for the indicators of whether a model used Beta-PERT or uniform distributions, but it might be more helpful to simply look at swapping out each distribution in turn and noting the relative change in mean and variance of your global conclusions. (You could do the same by simply increasing the variance a fixed amount in each Beta-PERT as well). The sensitivity analysis that you do provide suggests indirectly that these are not likely to be large effects, but currently they test the robustness of Sobol indices, rather than the model's direct output, to your modeling assumptions.

Referee #3 (Remarks to the Author):

This is my second time reviewing this paper. I was Reviewer #3 of the earlier version.

I am quite satisfied with the author's revisions to address my comments and those of the other reviewers.

I think this is an important paper, both scientifically and in the context of ongoing international negotiations for a plastics treaty, and I would like to see it published in a high-profile journal like Nature.

-Matthew MacLeod, Stockholm University

+++++

[Editorial note: The original referee #2 did not submit a report on the revised version of the paper. Therefore, referee #3 was asked to assess the authors' responses to the points raised by referee #2 in the previous round of review]

Comments of referee #3 on responses to concerns raised by referee #2

Referee #2:

Review of "A local-to-global emissions inventory of macroplastic pollution"

The research reported in this manuscript and the resulting macroplastic emission inventory represents novel and important contributions to the existing literature and data. The research uses existing municipality-level data of solid waste generation and management as training dataset for a machine learning algorithm (random forest) in order to derive a global, municipality-level dataset on plastic waste generation, management, and emissions. It does this using a very detailed material flow model, which enables the authors to provide unparalleled detail in the emission inventory (see extended data Fig. 1).

While the manuscript, with its 133 pages of SI, is challenging to review, all input data and methods appear to be sound and robust. The material flow model also looks sound.

Comments up to here are positive, and I believe Ref. #2 would support publication of the paper as the questions they raised below were (I think) mostly addressed by the authors...

My only concern is with the large amount of data required to populate it. My other concern is the size of the training dataset for the random forest. If I understand correctly, 691 municipal records, some duplicates and many doubtlessly incomplete, were used to generate datasets for 50,702 municipalities. Even discounting duplication, incompleteness, and representativeness of the existing records, that is a coverage of less than 2%. What can be said about the representativeness of the training dataset (input data) and thus the credibility of the results (output data)?

The reviewer's concern about low data coverage is not something the authors can do anything about. As the authors write in their rebuttal, there is data only for 2% of municipalities that represent 12% of the global population, and they extrapolate this data to the entire population. Of course there are uncertainties when this is done, but the authors have extensively documented their methods and they have included more caveats in the revised paper.

In my opinion it is good to make these extrapolations, and the empirical data basis is sufficient to make them meaningful! The alternative to extrapolation is to wait forever for "enough data" and in the meantime have no quantitative estimates at all at the global scale...

The reviewer's other criticism about a lack of clear constraints on the uncertainties in the emission estimates was also my main criticism of the paper. The uncertainties themselves are not something the authors can "fix", but my recollection of their revisions is that they did a much better job in the revised paper of clearly communicating the types and scale of uncertainties associated with their estimates, which is all they can do!

Here are some additional specific questions I have about the manuscript, the methods, and the data:

- Given the considerable uncertainties of all efforts to quantify mismanaged, leaked or emitted plastic waste, the reported results appear to be roughly in line with the existing literature; whereas the authors state that they are not.

The authors should try to provide a clearer analysis of similarities and discrepancies with the existing literature.

Again, I recall also raising a similar point in my review, and it is even referenced by the authors in their reply to this comment from Rev. 2. In the original paper the authors had highlighted what they saw as differences between their estimates and previous estimates which were not as comprehensive in scale and detail. In the revised paper they rewrote the relevant section of the paper in way more clearly compares their work to other estimates.

- It is probably buried somewhere in the 133 page SI, but it is unclear to me 1) which input data have probability density function(pdf), 2) where the pdfs come from, and 3) whether the pdfs are from observations or assumed. This should be explained somewhere where it can be easily found.

The author's response to this comment makes it clear that the data the Reviewer was looking for was included in the (extensive!) Supporting Information. The authors have made edits to the introduction to the relevant section of the SI to point to these data in particular, which seems like a good response to me...

- The information contained in extended data Fig.1 needs to be provided earlier, since it is necessary to avoid confusion when reading the results sections.

The authors made extensive edits to clarify terms (especially "open burning") which I think addresses this point.

- The observation that almost all of the plastic waste going to mismanaged disposal is burned is striking. What actual evidence is this based on, and what is the uncertainty of this result? E.g., are the disposal open burning values for Kuwait and Saudi Arabia based actual data or generated by the machine learning algorithm? What about open dumps of apparel waste? Is apparel and other non-packaging plastic waste included in this study?

The author's rebuttal of this comment makes it clear that there was a misunderstanding on the part of the Reviewer about the importance of burning. It is not the case that almost all plastic going to unmanaged disposal is burned, but rather that the majority of *emissions* from unmanaged disposal come from burning.

And, the burning values from Kuwait and Saudi Arabia that the Reviewer references were removed by the authors in response to a comment from Reviewer #1. So I think this comment has been well addressed.

And, the authors added a sentence explicitly clarifying that apparel (textiles) are excluded from the estimates.

- Which input datasets report uncollected plastic waste, and how much uncertainty is there for these values? It would seem to me that uncollected waste is not directly observable. Is this data inferred from other input data? This requires clarification.
- I would equally assume that the fraction of uncollected waste that is burned is not directly observable. Again, what is the source of this data, what is its global coverage, and what is its uncertainty? How clear or blurred is the line between uncollected open burned and disposal open burned?
- One more comment regarding the emission type open burning: I would assume that this process involves highly incomplete combustion, to the point of having a significant fraction of plastic waste either remaining or only partially combusted. What is known about this and should this be integrated into the model?

These three comments are all about how "burning" was described and quantified in the paper. I find it difficult to parse the reviewer's comments and the author's responses here since it is apparent that there was confusion on the part of both Reviewer #2 and myself about the author's definition and use of the term "burning".

What I see in the author's response (and in the revised paper) is that they made edits in the paper (or in some cases already had explanations in the extensive SI that could have been missed by the Reviewer and/or myself) that explain that there are two types of "burning" in the analysis and that it is an important process that requires further study to better constrain.

I think this is OK!

Author Rebuttals to First Revision:

Response to Reviewer 1

Comment 1

I would like to commend the authors on the work they put into this revision. My most substantial concern -- dependence between model outputs -- has been addressed in a reasonable manner.

We are pleased to hear you are satisfied with how we addressed this dependence and appreciate your guidance on the matter.

I dislike the terms epistemic and aleatoric uncertainty, partly because they're frankly pretentious but mostly because their mapping onto "uncertainty about model predictions" and "additional uncertainty about the values of a new data point" doesn't really match their original definitions. This does appear to be their current use, however, and instead I'll confine myself to noting that, in contrast to the discussion on p. 101 of the supplement, correlation between municipalities can also be induced due to epistemic uncertainty about modeling waste collection processes.

Thank you for this valuable observation. We interpret this comment as a suggestion that epistemic uncertainty also extends to uncertainty around how we model various processes (e.g. waste collection), rather than just uncertainty due to lack of data as we currently largely frame the discussion. We agree that this is an important form of uncertainty in our results, as it is in all modelling work, but also note that it is a very challenging form of uncertainty to quantify. We suggest some of this uncertainty is partly addressed in the improved sensitivity analysis, which assesses how sensitive model outputs are to assumptions we make on input distribution types. However, based on your comment, we have decided to include further discussion on this type of uncertainty to be more transparent, as shown on page 100 of the SI, and below:

“There are two types of uncertainty in our model: (1) Aleatoric uncertainty - observable differences in solid waste management practices between municipalities that may or may not have similar socio-economic characteristics (real world variability); and (2) Epistemic uncertainty – due to lack of knowledge that could in principle be known (e.g. lack of data to train the model for a particular set of geographical, socio-economic, or political conditions). In practice, our uncertainty quantification incorporates both types of uncertainty. We have discussed the methods for uncertainty quantification throughout the supplementary material, however we present a summary in this section for ease of reference. Additionally, we note that epistemic uncertainty does not just occur due to lack of data, but also because our model requires simplification of complex real-world phenomena as part of its formulation process. Whilst we attempt to address model assumptions as part of the sensitivity analysis (Section S.10), some epistemic uncertainty undoubtedly remains unquantified.”

My purpose in this bit of nit-picking is to query the authors choices not to include uncertainty about controlled dispersal of MSW (Supplement, In 2022, pg 93). This may be bimodal, but quantile random forests including both 0% and 100% for a given municipality is surely a reflection of epistemic uncertainty. If you claim to be accounting for both sources of uncertainty is it reasonable to leave this out? You might claim that this is a failure of the model in that it does not correspond to what we know about municipalities where we have reliable data, but then why should I trust it for municipalities that I'm simulating?

We recognise that our application of the quantile regression random forest for the uncontrolled disposal process did not predict as well as for other dependent variables. This is because the result of the quantile regression random forest is the full conditional distribution of leaf nodes, which will inevitably include both 0% and 100% controlled disposal predictions due to the binary nature of the data, meaning that the central prediction often moved away from the bounds. This situation is unlikely to be observed because in most real-world cases, as a municipality is unlikely to have more than one land disposal site, which would either be controlled or uncontrolled.

Ideally, we would have quantified the uncertainty in predictions of controlled disposal directly from the quantile regression random forest as we did for the other variables; however, we found that doing so did not predict well. This left us with a choice:

1. Use the quantile regression random forest model predictions and uncertainty, but achieve poor central predictions that do not accurately represent the test data or what we would expect to observe in reality
2. Use the majority estimate (as done in a standard application of random forest for classification), which achieves much better prediction of the test data and our observation of reality, but does not include uncertainty

In **Option 1**, the median result matched the test data in 63% of municipalities, whereas the results of **Option 2** matched 82% of the test data. In addition to the higher agreement with the test data, **Option 2** also provided, a clear binary result. We judged

that even though we were not able to incorporate uncertainty into the results of **Option 2**, that it would provide a result closer to reality than **Option 1**.

We are grateful to have a reviewer who understands our model sufficiently to have identified this as a weakness. We also acknowledge that others may query this decision too. Therefore, based on the reviewer's comment, we have now included the following explanatory narrative in the methods section of the main manuscript to highlight the limitations of our model and to be transparent about the predictive capability of our approach:

"We caution readers to consider the full uncertainty in our MFA results, particularly for municipal scale outputs where the ranges are generally much larger than national or regional scale aggregations. The origins of uncertainty in our model are discussed at length in Supplementary Information **Section S.9.2.2**. We also explain in **Section S.9.1.1** a specific circumstance where we decided not to quantify uncertainty for the uncontrolled disposal coefficient (tC3) due to limitations of the quantile regression random forest predictive capability for that particular aspect of the system. "

We have also included the following text in the supplementary material (pages 92-93) which expands on this point.

"The absence of uncertainty quantification for controlled disposal of MSW (tC3) primary input variable means that readers should treat this result with caution and not interpret our lack of uncertainty as conviction of the result. In particular, results for specific municipalities in LIC and LMC may occasionally be misclassified as uncontrolled (Error! Reference source not found.). We postulate this occurs in municipalities that have received official development finance for construction of a sanitary landfill, and therefore have controlled disposal levels beyond what the socioeconomics of that municipality would usually suggest. Whether these sites continue operating in controlled manner or revert to uncontrolled techniques is also unclear, as are the very definitions of 'controlled' in many of our sources of primary data input (**Section** Error! Reference source not found.). Methods such as the UN-Habitat6 Waste Wise Cities Tool (WaCT) should reduce this uncertainty in future data collection efforts by using standardised definitions, however, at present 'controlled' often has to be taken at face value.

Caution is also advised for results of controlled disposal in municipalities of UMC's as these are often at the point in their development where they transition to environmentally sound practices, and therefore, the UMC income category demonstrates the largest variation in levels of controlled disposal. Whilst this variation makes predictions challenging, the results match the test data reasonably well (Error! Reference source not found.). Confidence in the

controlled disposal result for municipalities in HIC's is also high given all eight cases of the test data were successfully predicted."

Table S 1. Summary of test dataset cases misclassified for the controlled disposal (tC3) variable when using the majority result of random forest predictions

Income category	Number of test dataset cases	Number of test dataset cases misclassified	Percentage of test dataset cases misclassified
HIC	8	0	0.0%
UMC	16	2	12.5%
LMC	33	5	15.2%
LIC	15	2	13.3%

Comment 2

A second comment is that it is helpful to be precise about the purposes of your sensitivity analysis and how those maps onto how you conducted it. The original analysis used Sobol indices to examine the impact of inputs on outputs, with the goal of understanding to what extent poorly-measured quantities were associated with variability in outputs.

The additional sensitivity analysis asks how the sensitivity analysis changes if we replace Beta-PERT distributions with uniforms. This is helpful in that it does not change what we think is important to measure better. That said, my original question had been about how your modeling choices impact the point estimates and uncertainty of your practical estimates -- does the act of changing one Beta-PERT to uniform change your estimates for total plastic waste, for example? To do this you could look at Sobol indices for the indicators of whether a model used Beta-PERT or uniform distributions, but it might be more helpful to simply look at swapping out each distribution in turn and noting the relative change in mean and variance of your global conclusions. (You could do the same by simply increasing the variance a fixed amount in each Beta-PERT as well). The sensitivity analysis that you do provide suggests indirectly that these are not likely to be large effects, but currently they test the robustness of Sobol indices, rather than the model's direct output, to your modeling assumptions.

Apologies for misunderstanding your previous comment, we believe your point is now clear to us. We have updated the results in the sensitivity analysis section (**Section S.10.2**) to also show how our main model output of plastic emissions (debris, burnt and total) vary between the second sensitivity analysis results (termed here 'counterfactual'

results) where uniform distributions are used compared to our main 'baseline' model using best estimates of PDFs. This is repeated for both globally aggregated results and by income categories.

As discussed on pages 107 and summarized in **Table S41**, there is an increase in emissions by approximately 30% for the counterfactual results, with open burning at uncontrolled dumpsites the primary driver of this. We suggest that this increase is reasonably minor considering all our major assumptions were removed, and note that none of our main conclusions were affected. Likewise, whilst plastic emissions from HIC's were predicted to increase by a large amount relative to the baseline results, these are still negligible in comparison to other income categories; therefore, this also does not change our conclusions. As such, we conclude that the model is indeed insensitive to any modelling assumptions we have made, whilst noting again the caveat that is uncertainty around controlled disposal predictions.

Response to Reviewer 3

We thank the reviewer for time and effort in reviewing our manuscript, and in the absence of additional questions, are pleased that they are satisfied with our changes and responses.